# *Data Mixture Inference:* What do BPE Tokenizers Reveal about their Training Data?

*Jonathan Hayase♡   *Alisa Liu♡   Yejin Choi♡♣   Sewoong Oh♡   Noah A. Smith♡♣

♡University of Washington   ♣Allen Institute for AI
{jhayase,alisaliu}@cs.washington.edu

## Abstract

The pretraining data of today's strongest language models is opaque; in particular, little is known about the proportions of various domains or languages represented. In this work, we tackle a task which we call *data mixture inference*, which aims to uncover the distributional make-up of training data. We introduce a novel attack based on a previously overlooked source of information: byte-pair encoding (BPE) tokenizers, used by the vast majority of modern language models. Our key insight is that the ordered list of merge rules learned by a BPE tokenizer naturally reveals information about the token frequencies in its training data. Given a tokenizer's merge list along with example data for each category of interest, we formulate a linear program that solves for the proportion of each category in the tokenizer's training set. In controlled experiments, we show that our attack recovers mixture ratios with high precision for tokenizers trained on known mixtures of natural languages, programming languages, and data sources. We then apply our approach to off-the-shelf tokenizers released with recent LMs. We confirm much publicly disclosed information about these models, and also make several new inferences: GPT-4O and MISTRAL NEMO's tokenizers are much more multilingual than their predecessors, training on 39% and 47% non-English language data, respectively; LLAMA 3 extends GPT-3.5's tokenizer primarily for multilingual (48%) use; GPT-3.5's and CLAUDE's tokenizers are trained on predominantly code (∼60%). We hope our work sheds light on current design practices for pretraining data, and inspires continued research into data mixture inference for LMs.[1]

## 1   Introduction

Pretraining data is at the heart of language model development, yet it remains a trade secret for today's strongest models. While it has become more common for model-producing organizations to release model parameters, they rarely share the pretraining data or important details about its construction. In particular, little is known about the proportion of different languages, code, or data sources present in the data; these design decisions require extensive experimentation that few organizations have the resources to perform, and have a significant impact on the resulting LM [7, 39, 35, 38, 49, 59].

While a long line of membership inference attacks [15, 51, 43, 52, 16, 21] aim to reveal information about the model's pretraining data, they typically focus on testing whether particular instances or authors contributed to the data. In this work, we tackle a different task we call *data mixture inference* which, given a set of disjoint categories that cover the training data (e.g., the set of natural and programming languages), aims to uncover each of their proportions.

---

[*]Equal contribution, ordered alphabetically.

[1]Code and detailed inferences available at `https://github.com/alisawuffles/tokenizer-attack`.

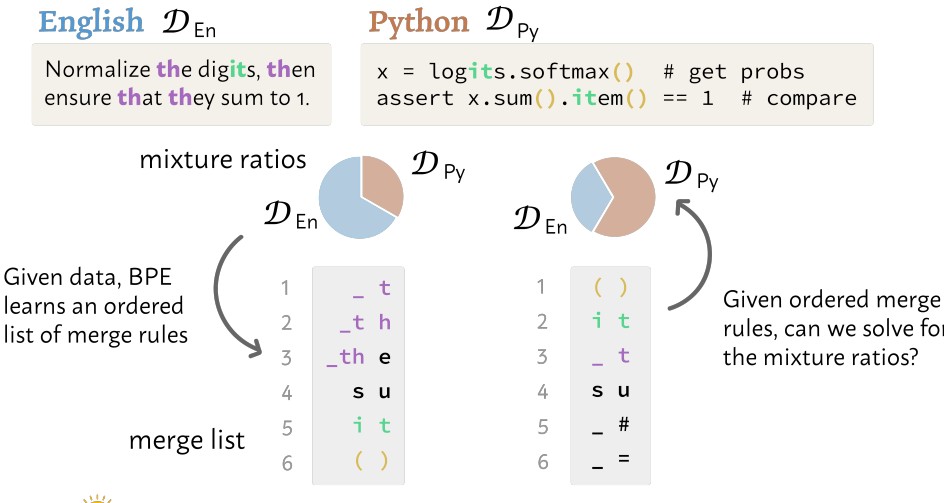

The learned merge list is sensitive to the mixture ratios!

Figure 1: **Illustration of our problem statement** on a simple example where two tokenizers are trained on different mixtures of English and Python data. During training, the BPE algorithm iteratively finds the pair of tokens with the highest frequency in the training data, adds it to the merge list, then applies it to the dataset before finding the next highest-frequency pair. To encode text at inference time, the learned merge rules are applied in order. The resulting order of merge rules is extremely sensitive to the proportion of different data categories present. Our goal is to solve for these proportions, a task which we call *data mixture inference*.

To this end, we identify a previously overlooked source of information: trained byte-pair encoding tokenizers (BPE; [50]), which are the near-universal choice for modern language models. Our key insight is that the *ordered merge rules* learned by a BPE tokenizer *naturally reveal information about the frequency of tokens* in the tokenizer's training data. During training, the BPE algorithm iteratively finds the ordered pair of tokens with the highest frequency, adds it to the merge list, and applies the merge to the dataset. Therefore, if the pair (;, \n) was merged in the 52$^{nd}$ step (as is the case for GPT-4O), then it must be the most frequent pair in the data after applying the preceding 51 merges; in this case, it is a signature of substantial code data. Note that at inference-time, new text is tokenized by applying the learned merge rules in-order. Open-source models require open tokenizers; even closed models often have open tokenizers for the purpose of estimating query cost.

Our method builds a linear program where the constraints are derived from the true most-frequent merge at every step in the merge list, and solves for the proportions of each category. We first demonstrate its effectiveness in controlled experiments where we train tokenizers on known mixtures of data. We consider three kinds of data mixtures: natural languages, programming languages, and data sources. Our method is highly effective, achieving accuracy between two and five *orders of magnitude* better than baselines based on tokenizer efficiency or inspection of the vocabulary.

Then, we apply our method to infer previously unknown distributional information about off-the-shelf, commercial tokenizers (the top of these merge lists are shown in §C.2 for qualitative inspection). We consider all tokenizers released with GPT, LLAMA, and MISTRAL model families, as well as GPT-NEOX, GEMMA, and CLAUDE, which we will refer to later using their associated model names. We corroborate reported information and public intuition about these tokenizers with exact numbers — GPT-2 is trained on predominantly English (99%), GPT-3.5 is the first in the GPT family to be trained extensively on code (63%), and LLAMA trains on only languages that use Latin or Cyrillic scripts. We also make several new inferences: GPT-4O is much more multilingual than its predecessors, training on 39% non-English text, with 68 languages that make up at least 0.1% of the data. LLAMA 3 extends GPT-3.5's tokenizer primarily for multilingual use, using 48% non-English text (and 30% code data). Finally, we surprisingly infer that all tokenizers we study are trained on 7% – 26% book data, potentially because books use a more standard vocabulary compared to the web.

Inferring tokenizer training data mixtures has several important implications. Ideally, the tokenizer training data is representative of the LM's pretraining data [58]; disconnects can lead to poor encoding

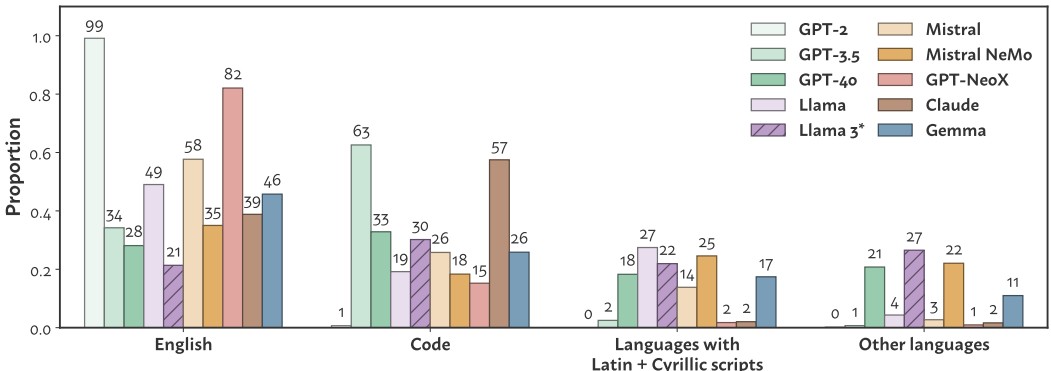

Figure 2: **Training data mixture predictions for several commercial tokenizers.** Complete results over 112 languages and 5 domains are given in §C; categories are grouped here for readability. We confirm that GPT-2 was trained overwhelmingly on English (99%), while GPT-3.5 is the first model in the GPT series to train on substantial code data (63%). GPT-4O is much more multilingual than its predecessors, with 39% of its corpus being non-English text. LLAMA is also multilingual, but focuses on languages using Latin or Cyrillic scripts (note this category in the figure excludes English). LLAMA 3* results are only based on the last 27,744 merges (the first 100K are copied from GPT-3.5), which we observe was primarily for multilingual adaptation.

of the pretraining text [2] and potential for "glitch tokens" that trigger degenerate model behavior [48, 30, 33]. Additionally, the tokenizer can be seen as a leading indicator of the model developer's priorities, as it is often designed to accommodate future models. Tokenizer training mixtures may also be used to accelerate model-based attacks, for instance by suggesting data categories to prioritize for membership inference. Finally, it can enable external auditing of training data for biases, by identifying under-represented languages or data sources.

## 2 Background: BPE tokenizers

Byte-pair encoding (BPE), introduced by Sennrich et al. [50] for NLP,[2] is a tokenization algorithm that learns subword-based encodings from training data. At a high level, the algorithm iteratively merges frequently co-occurring pairs of tokens until the desired vocabulary size is reached.

More precisely, the training text is first *pretokenized* by splitting it into "words" that limit the extent of tokenization. Merges cannot bridge these words, and thus the final learned tokens will be *parts* of these words. Pretokenization can be as simple as splitting on whitespace, so that common sequences of words (e.g., "*it is*") do not become a single token.

After pretokenization, the words are split into bytes, which form the starting vocabulary. Then, the BPE algorithm iteratively counts the frequency of each neighboring pair of tokens and picks the most frequent to be merged next. This merge is added to the merge list and applied to the entire text, and the merged token is added to the vocabulary. For instance, if the merge is (th, e), then all instances of the token sequence th, e will be replaced with the, which is added to the vocabulary. BPE then updates the frequencies of all pairs, and identifies the next most frequent. This continues until the desired vocabulary size is reached. At the end of training, the algorithm has learned an ordered list of merge rules $m^{(1)}, \ldots, m^{(M)}$.

To tokenize new text, the tokenizer splits the text into bytes and applies the learned merge rules, in order. As we will see, the merge list reflects rich distributional information about the training data.

## 3 Data mixture inference attack

Suppose we have a set of $n$ data categories of interest, and data distributions $\{\mathcal{D}_i\}_{i=1}^n$ for each one. Then suppose we receive a BPE tokenizer, which was trained on a large sample of text from the mixture $\sum_{i=1}^n \alpha_i^* \mathcal{D}_i$ with non-negative weights $\alpha^* \in \mathbb{R}^n$ satisfying $\sum_{i=1}^n \alpha_i^* = 1$. Given corpora

---

[2]Though, it originated in 1994 in the field of data compression [27].

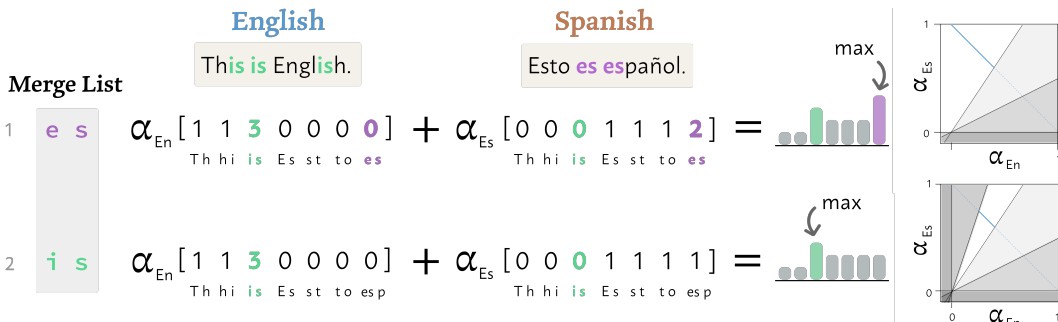

Figure 3: **Illustration of our method** on a simple example. We know that after applying in the first $t-1$ merges to the training data, the $t^{\text{th}}$ merge must be the most common pair. More explicitly, this means that $\alpha_i$ should give a vector in which the value corresponding to the true next merge is the maximum. Our attack collects these inequalities at every time step to construct the linear program.[3]

$\{D_i\}_{i=1}^n$ sampled from each of the $\mathcal{D}_i$ respectively, the goal of *data mixture inference* is to produce a good estimate $\hat{\alpha}$ of $\alpha^*$.

Now we describe how to set up the set of constraints that make up a linear program whose solution is this estimate (§3.1), reduce the storage requirements (§3.2), and improve efficiency (§3.3, §3.4).

### 3.1 Data mixture inference via linear programming

We build a linear program (LP) with variables $\alpha$ and constraints derived using information from the tokenizer and our sample corpora. The given tokenizer can be represented by an ordered list of merge rules $m^{(1)}, \ldots, m^{(M)}$. For each time step $t \in [M]$, we apply all preceding merge rules $m^{(1)}, \ldots, m^{(t-1)}$ to our corpora $D_i$ and use $c_{i,p}^{(t)}$ to denote how many times the token pair $p$ occurred in the partially merged text. We know that when the tokenizer was trained, the pair $m^{(t)}$ was more frequent at time $t$ than any other pair. In other words,

$$\sum_{i=1}^n \alpha_i c_{i,m^{(t)}}^{(t)} \geq \sum_{i=1}^n \alpha_i c_{i,p}^{(t)} \qquad \text{for all } p \neq m^{(t)}.$$

Collecting these constraints for all $t$ and $p$ defines a set of possible $\alpha$'s.

Of course, because we only have samples from the category distributions and not the exact data the tokenizer was trained on, the above linear program may not be feasible, as the counts will include some sampling noise. To address this, we relax the constraints by introducing new non-negative variables $v^{(t)}$ for all $t \in [M]$, and $v_p$ for all pairs $p$, which represent the degree of constraint violation for each merge and pair, respectively. We replace our constraints with new ones of the form

$$v^{(t)} + v_p + \sum_{i=1}^n \alpha_i c_{i,m^{(t)}}^{(t)} \geq \sum_{i=1}^n \alpha_i c_{i,p}^{(t)} \qquad \text{for all } p \neq m^{(t)}.$$

In general, we expect $v^{(t)}$ to be large when $m^{(t)}$ is over-represented in the tokenizer training data and $v_p$ to be large when $p$ is over-represented in the mixture defined by $\alpha$. This new system of constraints is guaranteed to be feasible, as the $v$'s can be made arbitrarily large. To produce the best possible estimate, our objective is to minimize the total constraint violation $\sum_{t=1}^M v^{(t)} + \sum_p v_p$. We call the resulting linear program LP1. To estimate $\alpha$, we solve LP1 and report the optimal value of $\alpha$ as $\hat{\alpha}$.

As written, LP1 can be prohibitively large. If our vocabulary has size $V$, the total number of constraints scales like $O(V^3)$ since there are $O(V)$ time steps $t$ to consider and $O(V^2)$ competing byte pairs $p \neq m^{(t)}$. Additionally, there are $O(V^2)$ variables $v_p$. The first step to reduce the size is to limit $t$ to the first $T$ merges. We will call this truncated program LP1$_T$. However, even for modest choices of $T$, LP1$_T$ can still have millions of variables and tens of billions of constraints. In the

---

[3]Technically, the elements of the vectors should be normalized by size of the language data, but in this case they are the same for the two languages so we show the unnormalized counts for readability.

following sections, we will describe how to efficiently solve $\texttt{LP1}_T$ using simultaneous delayed row (constraint) and column (variable) generation [12, 23].

## 3.2 Efficient storage of pair counts

First, as a preprocessing step, we apply the target tokenizer to each language corpus $D_i$, recording the pair counts $c_{i,p}^{(t)}$ after each merge is applied for later use. Naively, this would require a large amount of space, since the number of possible pairs $p$ scales like $O(V^2)$. However, note that $c_{i,p}^{(t)} \neq c_{i,p}^{(t+1)}$ only when $p$ overlaps with $m^{(t)}$. In other words, pairs with no overlap with the most recent merge will have unchanged counts. Thus, there are only $O(V)$ differences between $c_{i,\cdot}^{(t)}$ and $c_{i,\cdot}^{(t+1)}$. In practice, the number of changes caused by a single merge is usually a few hundred at most. By saving only the incremental changes from each set of pair counts to the next, we can efficiently record the pair counts at every iteration of the tokenization process.

## 3.3 Efficient constraint violation detection

Our plan is to solve $\texttt{LP1}_T$ using only a subset of its constraints, giving a potential solution $(\alpha, v)$. We then check whether $(\alpha, v)$ violates any of the constraints of $\texttt{LP1}_T$ and add any such constraints to the subset. This requires an efficient method to detect violated constraints, which we describe below.

For convenience, let $s_p^{(t)} := \sum_{i=1}^n \alpha_i c_{i,p}^{(t)}$ and recall that, for a given time step $t$, we want to check whether $v^{(t)} + s_{m^{(t)}}^{(t)} \geq \max_p(s_p^{(t)} - v_p)$. Naively, we would do so by iterating over all possible $p \neq m^{(t)}$ to see if the constraint is violated, which can be quite costly. Moreover, we must do this for all $t \leq T$. However, by taking advantage of the structure of the $s_p^{(t)}$ as $t$ varies, we can reduce our work substantially.

The first step is to take each initial pair $p$ and add it to a priority queue with priority $s_p^{(0)} - v_p$. This can be done in aggregate in $O(V^2)$ time using a fast heap building algorithm. Now, we can iterate through the pairs in descending $s_p^{(0)} - v_p$ order using the queue's $\texttt{delete-min}$ operation. For each pair $p$, we can check whether $v^{(0)} + s_{m^{(0)}}^{(0)} > s_p^{(0)} - v_p$ and if not, we mark the corresponding constraint as violated. Once we find a $p_{\text{sat}}$ satisfying the constraint, we stop, since all pairs remaining in the queue must satisfy their constraints. If there were $k$ pairs before $p_{\text{sat}}$ in the queue, then the total time taken is $O(k \log V)$.

Crucially, we can quickly update the priority queue to reflect the state at $t = 1$. Since we precomputed all count changes from $c_{i,p}^{(0)}$ to $c_{i,p}^{(1)}$, we know what queue entries need to be updated or inserted. If $k_{\text{new}}$ new pairs were created and $k_{\text{old}}$ pairs had their counts changed when pair $m^{(0)}$ was merged, then we can update the priority queue using $k_{\text{new}}$ $\texttt{insert}$ operations and $k_{\text{old}}$ $\texttt{decrease-priority}$ operations, which can be done in $O((k_{\text{new}} + k_{\text{old}}) \log V)$ time. Now that we have updated the priority queue, we can repeat the above procedure to check for any constraint violations for $t = 1$. By iterating this process, we can quickly check for violated constraints for all $t \leq T$.

## 3.4 Lazy variable and constraint generation

Now we are ready to efficiently solve $\texttt{LP1}_T$. We begin by guessing uniform proportions for $\alpha$, and $v^{(t)} = v_p = 0$ for all $t, p$. Then we use our constraint checker to identify violated constraints of $\texttt{LP1}_T$ and construct a lazy version of $\texttt{LP1}_T$, which we denote $\texttt{LP2}_T$, using only those constraints and the variables they contain. We then solve $\texttt{LP2}_T$, which gives us a new guess for $\alpha$. We repeat the above steps, adding progressively more constraints (along with their variables) to $\texttt{LP2}_T$ until we find a solution that is also feasible for $\texttt{LP1}_T$. It follows that this solution is optimal for $\texttt{LP1}_T$ since the two programs share the same objective. This is guaranteed to happen eventually because there are a finite number of constraints and variables to add.

In practice, the constraint violation detection can typically check $T = 30000$ merges in less than 10 seconds. On difficult instances, such as those for commercial tokenizers in §5, the full solve can take up to a day to complete. Easier instances like those in Table 1 can be solved in a few minutes.

Table 1: **Experimental results for controlled experiments.** The settings we consider are mixtures of natural languages, mixtures of programming languages, and mixtures of domains. $n$ denotes the number of categories in the mixture, which are drawn from 112 natural languages, 37 programming languages, or 5 domains. In each cell, we report the mean and standard deviation of $\log_{10}(\text{MSE})$ over 100 trials; note that a decrease by 1 corresponds to a $10\times$ improvement in the MSE. In addition to a **Random**-guessing baseline, we implement two alternative approaches to the problem: **TEE** (Tokenizer Encoding Efficiency) uses the tokenizer's encoding efficiency on each data category, and **TC** (Token Classification) assigns each token in the vocabulary to a data category based on frequency.

| $n$ | Method | Languages | Code | Domains |
|---|---|---|---|---|
| 5 | Random | $-1.39_{\pm 0.36}$ | $-1.39_{\pm 0.36}$ | $-1.39_{\pm 0.36}$ |
| | TEE | $-2.02_{\pm 0.41}$ | $-2.54_{\pm 0.42}$ | $-1.69_{\pm 0.29}$ |
| | TC | $-2.12_{\pm 0.49}$ | $-1.92_{\pm 0.36}$ | $-1.64_{\pm 0.35}$ |
| | Ours | $\mathbf{-7.30}_{\pm 1.31}$ | $\mathbf{-6.46}_{\pm 0.79}$ | $\mathbf{-3.74}_{\pm 0.94}$ |
| 10 | Random | $-1.84_{\pm 0.23}$ | $-1.84_{\pm 0.23}$ | – |
| | TEE | $-2.29_{\pm 0.26}$ | $-2.59_{\pm 0.24}$ | – |
| | TC | $-2.55_{\pm 0.36}$ | $-2.38_{\pm 0.20}$ | – |
| | Ours | $\mathbf{-7.66}_{\pm 1.04}$ | $\mathbf{-6.30}_{\pm 0.64}$ | – |
| 30 | Random | $-2.70_{\pm 0.13}$ | $-2.70_{\pm 0.13}$ | – |
| | TEE | $-3.07_{\pm 0.16}$ | $-3.15_{\pm 0.13}$ | – |
| | TC | $-3.42_{\pm 0.23}$ | $-2.38_{\pm 0.20}$ | – |
| | Ours | $\mathbf{-7.73}_{\pm 1.12}$ | $\mathbf{-5.98}_{\pm 1.11}$ | – |
| 112 | Random | $-3.82_{\pm 0.07}$ | – | – |
| | TEE | $-4.15_{\pm 0.08}$ | – | – |
| | TC | $-4.46_{\pm 0.12}$ | – | – |
| | Ours | $\mathbf{-7.69}_{\pm 1.28}$ | – | – |

## 4   Experiments

In our initial experiments, we train tokenizers on known data mixtures and measure the accuracy of our attack's prediction. We consider mixtures of natural languages, programming languages, and data sources (which we also refer to as domains).

### 4.1   Setup

Because BPE tokenizers operate on bytes, we measure the proportion of each category in terms of bytes. Each tokenizer is trained on a mixture of $n$ categories, where $n$ varies from 5 and 112. We randomly sample the $n$ categories and their weights from the unit simplex (using the algorithm from [53]), and train 100 tokenizers on 10 GB of data. The data for each category is sampled from the corresponding corpus; if there is not enough data for any category (e.g., we have many low-resource languages), we duplicate the data until the necessary amount is achieved, to preserve the desired mixture ratio. We train tokenizers using the HuggingFace `tokenizers` library with a maximum vocabulary size of 30,000, and apply a minimal set of common pretokenization operations: we split on whitespace and only allow digits to be merged with other contiguous digits.

After training the tokenizers, we apply our attack. We estimate merge frequencies for each category by sampling 1 GB of data per category, or less if there is not that much data. Note that the data used for training the tokenizer and estimating pair frequencies are sampled from the same distribution, but are not necessarily the same data. We use **mean squared error** to evaluate the estimated proportions, $\text{MSE} := \frac{1}{n}\sum_{i=1}^{n}(\hat{\alpha}_i - \alpha_i^*)^2$. In practice, we report $\log_{10}(\text{MSE})$.

In §B.4, we analyze how our attack's performance varies with the amount of data used and the number of merges $T$ considered from the merge list.

**Natural Language Mixtures**   We use the Oscar v23.01 corpus [1], which is based on the Nov/Dec 2022 dump from Common Crawl. We consider the 112 languages with at least 1 MB of data.

**Programming Language Mixtures**    We use the GitHub split of RedPajama [22]. To determine the programming language for each record, we map the file extension to its associated language (e.g., `.py` → `Python`). This leads to a total of 37 programming languages.

**Domain Mixtures**    We consider the following five English domains (adapted from [38]), instantiated by data from the RedPajama dataset: **Wikipedia**, containing English Wikipedia dumps from Jun-Aug 2022, **Web**, Common Crawl data that was de-duplicated and filtered for English, **Books** from the Gutenberg Project and Books3 of The Pile, **Code** from GitHub, and **Academic**, which contains LaTeX files of scientific papers on ArXiv.

For dataset details, e.g., the full list of categories and data sizes, please see §B.

## 4.2   Baselines

We construct two intuitive baselines: one based on tokenizer encoding efficiency and one based on analysis of tokens in the vocabulary. Neither takes the BPE training algorithm into account.

**Baseline based on tokenizer encoding efficiency (TEE)**    Intuitively, we expect data categories with greater representation in the training data to be encoded more efficiently by the resulting tokenizer. To capture this, we calculate a given tokenizer's byte-to-token ratio on each category (a more efficient tokenizer will encode more bytes per token), then normalize it by that of a reference tokenizer trained on *only* that category, to control for different categories being inherently easier or harder to encode. Then we learn a log-log linear model to predict each category's true proportion given the encoding efficiency. To ensure correctness, we normalize the resulting set of predictions into a probability distribution.

**Baseline based on token classification (TC)**    We also consider a baseline that assigns each token in the vocabulary to a data category based on its empirical frequency in the sample data. Intuitively, we expect that if there is a large proportion of e.g., English data in the training data, then there will be more "English" tokens. For each token in the vocabulary, we count its occurrences in data sampled from each category, and assign it to the one in which it is most frequent (we find that hard assignment outperforms all variations of soft assignment). Then we count the number of tokens assigned to each category, and normalize the counts to produce an estimate of the data mixture.

## 4.3   Results

Shown in Table 1, our attack is highly effective. Over all mixture types and values of $n$, we achieve mean MSE two and six *orders of magnitude* better than random guessing. In contrast, the baselines based on tokenizer efficiency (TEE) and token classification (VC) do not come close to the kind of precision possible with our attack, achieving at best one order of magnitude better than random.

We observe that the setting with the highest attack success is mixed languages, whereas the most challenging is mixed English domains. This is perhaps unsurprising when considering the source of signal for our attack, which is the different token pair frequencies in different data categories. Intuitively, we would expect these to be very different for different natural languages, which have distinct vocabularies. In contrast, programming languages can share many syntactic features, such as using indents, curly brackets {}, and English variable names. Even more so, English data from different domains (e.g., books vs. Wikipedia) will largely share the same vocabulary but have subtle differences in token frequencies due to style, topic, and formatting. Nonetheless, even in this most challenging setting, we achieve accuracy $100\times$ better than either baseline.

## 5   Attacking commercial tokenizers

After validating our attack in synthetic experiments (§4), we apply it to infer training data mixtures of off-the-shelf commercial tokenizers. We refer to tokenizers by the name of the model they were first released with, whose pretraining data they most likely reflect. We consider GPT-2 [47], GPT-3.5 [45], GPT-4O [46], LLAMA [56], LLAMA 3 [37], MISTRAL [4], MISTRAL-NEMO [6], GPT-NEOX [13], CLAUDE [8], and GEMMA [55]. While many of these are closed models, their tokenizers are publicly available so that customers can estimate the cost of queries ahead of time. We note that the

LLAMA, GEMMA, and MISTRAL tokenizers use *characters* instead of bytes as the base vocabulary for the BPE algorithm; this does not affect our attack, but we discuss the distinction in §C.6.

In these experiments, we aim to infer the proportion of different natural languages, code, and English domains, for a total of 116 categories. We consider code as a single category (not split into separate programming languages) because some languages like `Markdown` and `Pod6` are almost entirely English, and we do not expect the distribution of programming languages in pretraining to differ substantially from that of GitHub, the largest public code hosting platform. To infer the distribution of English domains, we replace the English category with the four English domains from §4 (web, books, Wikipedia, and academic), which we expect to approximately cover the English data.

Our predictions are shown in Figure 2, with specific numbers in §C.1. Below, we discuss our findings in comparison with publicly disclosed information about these models.

## 5.1  GPT models

All tokenizers accompanying GPT models are open-source on `tiktoken`.[4] There are three such tokenizers, released with GPT-2, GPT-3.5, and the very recent GPT-4O.[5]

**GPT-2**    GPT-2 was trained on WebText, consisting of text scraped from outbound links from Reddit, and filtered to be English-only. The GPT-2 tokenizer was reused for GPT-3. Indeed, we confirm the training data consists of 99.1% English. However, we surprisingly estimate that only 83.6% of the data was web, with another 15.4% being books, which were not explicitly included in WebText. In a data contamination analysis, the authors indeed report that they find books in WebText, but our estimate suggests the contamination may be deeper. We note that books were a popular source of pretraining data for early Transformer LMs, with GPT-1 being trained entirely on BooksCorpus [63].

**GPT-3.5**    The GPT-3.5 family of models is known to depart from its predecessors by training on large amounts of code: the first model in this family was `code-davinci-002`, trained on text and code. In fact, some evidence suggests that GPT-3.5's large leap in reasoning abilities comes from this code data, which intuitively requires similar procedural skills [26]. The GPT-3.5 tokenizer was reused for GPT-4.

Indeed, we estimate that GPT-3.5 is trained on 62.6% code. In the domain breakdown, 27.3% is of the data is web, 6.8% books, and 0.2% academic articles. The substantial representation of books (though lower than GPT-2) is consistent with findings that this model has memorized a wide collection of copyrighted books [20].

**GPT-4O**    GPT-4O is a multimodal model announced as more multilingual than its predecessors; its tokenizer achieves a better compression rate on non-English languages, and the model has notably better non-English performance.

Our findings support this. GPT-4O is trained on 39.0% non-English text, compared to only 3.2% for GPT-3.5. The language distribution has a thick non-English tail, with 68 languages that make up at least 0.1% of the data: the most common are French (2.9%), Russian (2.8%), Spanish (2.8%), Portuguese (2.3%), Dutch (2.0%), German (1.8%), Arabic (1.6%), and Hindi (1.4%). Additionally, GPT-4O was trained on 7.4% books.

## 5.2  LLAMA MODELS

**LLAMA**    The training data for LLAMA is known to be primarily English, though the Wikipedia split "*covers 20 languages which use either the Latin or Cyrillic scripts:* `bg, ca, cs, da, de, en, es, fr, hr, hu, it, nl, pl, pt, ro, ru, sl, sr, sv, uk`". The training data is reportedly sourced from Common Crawl (67% of examples), C4 (15.0%), Github (4.5%), Wikipedia (4.5%), Books (4.5%), ArXiv (2.5%), and StackExchange (2.0%). The LLAMA tokenizer was reused for LLAMA2.

---

[4]`https://github.com/openai/tiktoken/blob/main/tiktoken/model.py`

[5]Technically GPT-3's tokenizer has a distinct identifier from that of GPT-2, but it differs only in 24 extra tokens at the end of the vocabulary; these tokens are made up entirely of spaces. Indeed, the GPT-3 technical report states that it "*reus[ed] the tokenizer of* GPT-2."

We corroborate that LLAMA is indeed primarily made up of the stated languages; when combined with code, this sums to 95.7% of the training corpus. Indeed, other generally high-resource languages, such as Chinese, Arabic, and Hindi, have 0.0% representation. However, we predict a very different domain distribution compared to what is reported in the paper for LLAMA's pretraining data. We predict that the tokenizer is trained on 23.1% books, 11.3% web, 6.7% Wikipedia, and 8% ArXiv. The high representation of books is surprising – we hypothesize that the tokenizer was trained on a different distribution than the LM, primarily focusing on books which uses a more standard vocabulary compared to web data.

**LLAMA 3**   We observe that LLAMA 3, rather than training a new tokenizer, extends GPT-3.5's merge list (of 100,000 merges) with an extra 27,744 merges. Thus, we apply our attack to these new merges to infer what data was used to *extending* the LLAMA 3 tokenizer. It is reported that LLAMA 3 is trained on more than 5% of "*high-quality non-English text that covers over 30 languages*." We find that LLAMA 3 is extended with primarily non-English text (48.5%) and code (30.2%), indicating that the goal of extending GPT-3.5's tokenizer was primarily for multilingual use.

### 5.3   MISTRAL

**MISTRAL**   MISTRAL models "*handle English, French, Italian, German and Spanish*" [5]; indeed, we find that these are the top five languages in the training data. There is a long tail of other languages, but they predominantly (97%) use either the Latin or Cyrillic script.

**MISTRAL NEMO**   In contrast, MISTRAL NEMO was "*designed for global, multilingual applications, bringing frontier AI models to... all languages*." It introduces a new tokenizer (based on `tiktoken` instead of `sentencepiece`), which is the most multilingual of tokenizers we study, training on 46.6% non-English text. French (6.3%) and Arabic (4.8%) are the most common non-English languages.

### 5.4   GPT-NEOX

The tokenizer of GPT-NEOX [14] was trained on the Pile [29] with "*certain components... upsampled*." It is popularly re-used in open-source model development, including by OLMO [31], PYTHIA [13], and DCLM [34]. Though our domains do not map neatly onto the constituent datasets of the Pile, our inference is generally consistent, with a prediction of 43.7% web, 26.3% books, 12.1% academic, 15.2% code, and 2.7% non-English text.

### 5.5   GEMMA

GEMMA [55] is reported as training on "*primarily-English data from web documents, mathematics, and code*." Gemma uses a subset of the Gemini tokenizer; we believe it is likely that this subset was obtained by truncation, which would make our inferences valid for Gemini as well. We predict that GEMMA is trained on 45.7% English, which comes from 25.6% web, 12.8% books, 4.3% academic, and 3.0% Wikipedia. It is also trained on 28.4% non-English text, which explains its large multilingual vocabulary of 256,000 tokens. However, compared to GPT-4O, the multilingual representation is more skewed toward languages that use Latin or Cyrillic scripts.

### 5.6   CLAUDE

Very little is known about models from the CLAUDE family, but a remark in the Anthropic SDK suggests that CLAUDE 1 [8] and 2 [9] share the same tokenizer, which is open-source, while CLAUDE 3 [10] uses a different (closed) tokenizer. Our attack predicts that CLAUDE was trained on 57.5% code, 38.8% English, and 3.7% other languages. Moreover, half of its data (17.4% overall) comes from books, with substantial contribution from Wikipedia (3.7%) and academic text (5.1%) as well. The lack of multilingual training data likely explains why a new tokenizer was trained for CLAUDE 3, which boasts "*increased capabilities... in non-English languages*" [10].

# 6 Robustness analysis

## 6.1 Is the attack robust to distribution shift?

We measure the impact of using out-of-distribution data instead of data from the tokenizer's training distribution to count merge frequencies. Note that in the main experiments, we show that the attack is effective at separating English domains, so performance degradation under distribution shift is expected. To empirically measure this, we train tokenizers on mixtures of $n = 10$ languages using web data from Oscar, but estimate merge frequencies using corresponding language splits of Wikipedia, a substantially different domain. Using the same settings as the main experiments (§4), we achieve $\log$ MSE of $-3.53$, compared to $-7.66$ with no shift. Thus, the attack performance drops considerably under this extreme distribution shift, while remaining $100\times$ better than random.

## 6.2 Is the attack robust to unaccounted-for categories?

In a typical attack setting, the attacker may not have explicitly accounted for every source of data used to train the tokenizer. To show that our approach is robust in the presence of such data, we modify our $n = 112$ languages experiment from our main experiments (§4). We randomly withhold a random subset of 1 to 50 languages from the solver and measure the resulting prediction error on the remaining languages. Results are shown in Figure 4. Although the performance of our method does worsen as the amount of unknown data increases, the predictions remain substantially better than random.

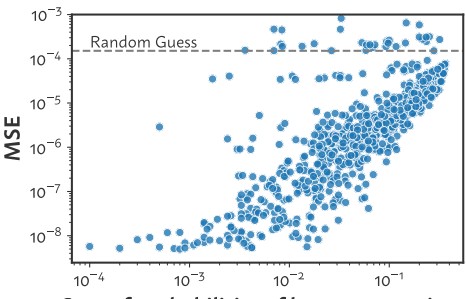

Figure 4: Performance remains much better than random even with large amounts of unknown data.

# 7 Related work

**Attacks on LMs**  Membership inference, the task of inferring whether a particular example was part of the training data, has been studied in great depth for language models [15, 17, 40, 42, 24, 51]. Attacks commonly use model-assigned probabilities, potentially with another reference model, to make an inference. The problem remains extremely difficult, with a recent survey finding that many attacks perform near random when pretraining data is deduplicated, as in common practice [24].

Closely related to this, some memorization attacks aim to extract memorized examples from the pretraining data via prompting attacks [18, 44]. Recently, there has even been progress in recovering the parameters of black-box LMs through model stealing attacks [25, 19].

**Distribution inference**  In contrast to membership inference, *distribution inference* is concerned with inferring global properties of a model's training data, but has not previously been studied for LM pretraining data. In early works, these attacks were successful against machine learning classifiers [11, 28], CNNs [54], and GANs [62]. More recently, some works show that multi-party machine learning can leak property information such as which authors contributed to the data [41, 61]. All previous distribution inference attacks take a meta-classifier approach, where models are trained on datasets with different properties, then a meta-classifier is trained using those models.

# 8 Conclusion

In this work, we present a data mixture inference attack that solves for the distributional make-up of a tokenizer's training data, which is commonly representative of the language model's pretraining data. Beyond the properties we study, we believe there is still a wealth of information hidden in tokenizer merge lists. This can shed light on the secretive and often contentious design decisions surrounding pretraining data today, potentially enabling external auditing for safety, copyright issues, and distributional biases. We hope our work will inspire continued research into inferring more global properties of training data, for tokenizers and language models more generally.

## Acknowledgments

We would like to thank Luca Soldaini for identifying the cause of redundant merges in some commercial LLM tokenizers (discussed in §C.3), and Jiacheng Liu, Orevaoghene Ahia, Xiaochuang Han, Muru Zhang, Thao Nguyen, Scott Geng, Rulin Shao, Zhaofeng Wu, and the greater UW NLP community for valuable feedback and conversations on this work. We are also grateful to the HuggingFace user `Xenova` for posting `tokenizers`-compatible versions of many tokenizers. This work is supported by Microsoft Grant for Customer Experience Innovation, the National Science Foundation under grant No. 2019844, 2112471, and 2229876 and DMS-2134012. Both co-first authors (JH and AL) are supported by the NSF Graduate Research Fellowship Program.

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

## A  Discussion of possible defenses

We discuss some possible approaches for defenses to our attack.

**Post-hoc changing the order of merge rules**  Model producers may consider changing the order of merge rules, which is the source of signal for our attack, after the tokenizer is trained. However, naively re-ordering merge rules for a tokenizer would be damaging, as it can lead to unfamiliar encodings of words as well as entirely unreachable tokens. The only functionally-equivalent re-ordering would be within contiguous sections of merge rules, where each token appears exclusively on the left or right side of all merges it appears in. In this case, we can easily adapt our method by working at the level of contiguous non-conflicting sections instead of individual merge rules, as we know that each section has higher frequencies than the next.

**Hiding pretokenization rules**  Our method relies on a reasonable reconstruction of the pretokenization rules, which control what kinds of merge rules are considered. It is not necessary to share pretokenization rules, as they are not strictly necessary for inference. However, we find that important pretokenization rules (like whether to pre-tokenize on spaces, digits, and punctuation) are easy to infer from manual inspection.

**Not using BPE tokenizers**  Model producers may choose to forgo BPE tokenizers entirely in future models. Despite the current popularity of BPE, there is a lively area of research into alternative methods of tokenization [57, 60, 36, 3]. While we only explore BPE tokenizers in this paper, it is plausible that any tokenizer learning algorithm will leak information about its training data, as they are specifically developed to best encode the given data.

# B   Experiment details & additional results

## B.1   Data details

The full set of categories that we use in §4 and the amount of data available for each category are shown in Table 5, Table 6, and Table 7.

**Language mixtures**   We use OSCAR-23.01, which is the January 2023 version of the OSCAR Corpus based on the November/December 2022 dump of Common Crawl. We only keep languages with at least 1 MB of data.

**Code mixtures**   We use the GitHub split of RedPajama-Data-1T, which is an open reproduction of LLAMA's training data.

**Domain mixtures**   We use five splits of RedPajama, namely Wikipedia, Common Crawl, Books, Github, and ArXiv. To reduce disk usage, we download only 8% of the CC URLs.

Below, we enumerate the licenses for these datasets.

- Oscar: CC0 1.0 Universal
- RedPajama has different licenses for each subset
    - C4: ODC-BY
    - GitHub: MIT, BSD, or Apache
    - Books3: MIT
    - Project Gutenberg: Apache 2.0
    - ArXiv: CCo 1.0
    - Wikipedia: CC-BY-SA-3.0

## B.2   Compute details

We run all of our experiments on CPUs. For training tokenizers and calculating pair frequencies, we use 16–32 CPUs and a variable amount of memory (ranging from 4 GB to 64 GB) depending on the data. Training a tokenizer on 10 GB of data (as in our experiments) usually takes around 10 minutes, while calculating pair counts takes between 1 minute and 2 hours, again depending on the data. To solve our linear programs, we use Gurobi [32].

## B.3   Baseline further discussion

For the baseline based on tokenizer efficiency, we plot the relationship between the true training proportion and tokenizer efficiency (as the normalized byte-to-token ratio) in Figure 5. As expected, the more data for a particular language there is in training, the more efficiently the tokenizer encodes that language. However, the correlation is clearly imprecise, with the true proportion ($x$-axis) varying up to an order of magnitude given the encoding efficiency ($y$-axis).

Between the baselines based on tokenizer encoding efficiency (TEE) versus vocabulary item categorization (VIC), we find that TEE performs better for mixtures of code, while VIC performs better for mixtures of languages. This makes sense, because different languages have very distinct vocabularies, while vast inherent differences between languages may make encoding efficiency hard to compare, even when using normalization.

## B.4   Scaling Analysis

Using our setup for controlled experiments (§4), we analyze how our attack's performance varies with the amount of data used (§B.4.1) and the number of merges we consider from the merge list (§B.4.2).

### B.4.1   How many data samples should we use from each category?

We explore how the attack's performance scales with the amount of data sampled from each distribution $\mathcal{D}_i$ for calculating pair counts. For each type of mixture considered in §4, we train 100

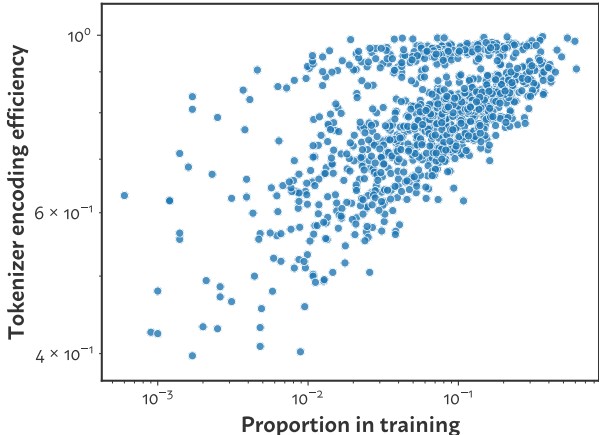

Figure 5: **Relationship between a language's proportion in training and the resulting tokenizer's encoding efficiency on that language**, shown for mixtures of $n = 10$ languages. The encoding efficiency is defined as the byte-to-token ratio of a given tokenizer on a given language, normalized by that of a tokenizer trained *only* on that language. While more training data leads to better encoding efficiency, the correlation is not strong enough to recover a prediction nearly as precise as our attack. A baseline based on this relationship achieves $\log_{10}$ MSE of $-2.22$, compared to our attack's $-7.66$.

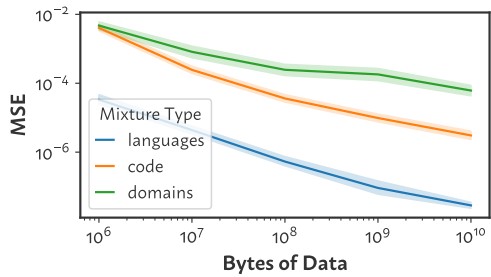

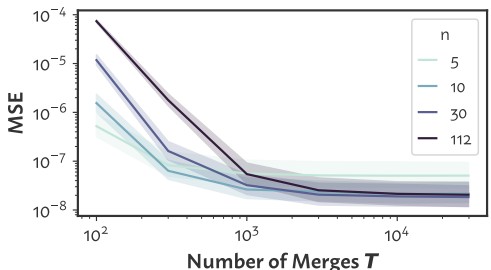

Figure 6: **Scaling the amount of data used for estimating pair frequencies** (§B.4.1), for mixtures of $n = 5$ categories. Sampling more data per category produces more precise inferences.

Figure 7: **Scaling the top $T$ merges used in the merge list** (§B.4.2). For mixtures of more categories (larger $n$), considering more merges (larger $T$) becomes more useful.

new tokenizers considering only categories with at least 10 GB of data available. For our attack, we compare sampling 1 MB, 10 MB, 100 MB, 1 GB, and 10 GB of data for each category, and use $T = 3000$ merges. Shown in Figure 6, more data consistently improves the accuracy of predictions.

### B.4.2 How many merges should we consider?

Next, we investigate how performance scales with the number of merges $T$ that we apply from the merge list. Using the same 100 tokenizers from §4, we solve for the data mixture using various choices of $T \in [30, 30000]$. Shown in Figure 7, we find that when there are more categories, it is useful to consider more merges. This makes sense because more constraints may be needed to bound the solutions in higher dimensions.

### B.4.3 Scaling analysis under distribution shift

Here, we repeat the distribution shift experiments in §6.1 while varying the amount of data and merges used. Shown in Figure 8 and Figure 9, we find a U-shaped curve for how performance scales with the amount of data used for calculating pair frequencies, unlike our main experiments in §4.

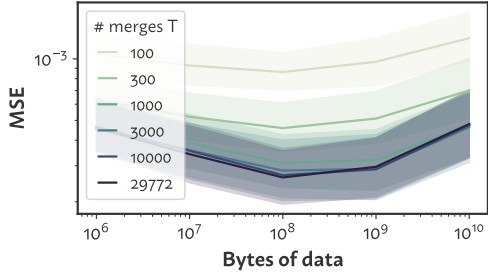 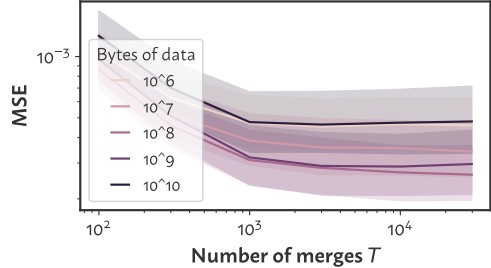

Figure 8: **Scaling the amount of data used for estimating pair frequencies** for distribution shift experiments from §6.1.

Figure 9: **Scaling the top $T$ merges used in the merge list** for distribution shift experiments from §6.1.

## C Commercial tokenizers

In this section, we provide more detailed results and discussion of commercial tokenizers.

### C.1 Full results for commercial tokenizers

We report the full inferences for commercial tokenizers (§5) over 116 categories (111 languages, 4 English domains, and code) in Table 4.

### C.2 Snapshot of commercial tokenizer merge lists

We show the first 50 merges of the commercial tokenizers we study in Table 2 for qualitative inspection.

### C.3 Handling redundant merges

We observe that the merge list of LLAMA, LLAMA 3, GEMMA, and MISTRAL contain clusters of redundant merge rules. For instance, in the LLAMA 3 merge list, we see the sequence of merges _ the, _t he, and _th e, as well as _ and, _a nd, and _an d. Because the merge path for every token is unique, it is impossible for more than one of these merges to ever be used, and we empirically verify this by applying the tokenizer to a large amount of text.

We find that this is an artifact of the conversion from sentencepiece to Huggingface tokenizers format. To construct the merge list, the conversion algorithm naively combines every pair of tokens in the vocabulary, and then sorts them by token ID, which represents order of creation. While this is functionally correct, because the redundant merges are not products of the BPE algorithm (i.e., they do not actually represent the most-likely next-merge), we need to remove them for our algorithm. To do this, we do some simple pre-processing: for every cluster of redundant merges, we record the path of merges that achieves each merge; the earliest path is the one that would be taken, so we keep that merge and remove the rest.

As an aside, this means that a tokenizer's merge list can be completely reconstructed from its vocabulary list *ordered by token creation*. Given only the resulting token at each time step, we can derive the corresponding merge.

### C.4 Manual merges in GEMMA

We notice that the first 1,395 merges in GEMMA consistent entirely of merges of consecutive \n, followed by merges of consecutive \t, and finally merges of consecutive whitespace characters _. They appear to be placed there manually and are not organically learned by the BPE algorithm, and do not correspond to increasing vocabulary IDs. Therefore, we remove these merges so that the remaining ordered merge rules align with monotonically increasing vocabulary ID.

We note that LLAMA and GEMMA, the two other tokenizers based on sentencepiece, also contain manually inserted merges of whitespace, but at the end of the merge list instead of the top. Since they are not within the top $T = 30,000$ merges we consider, we do not do any special pre-processing.

### C.5 GPT tokenizers

While GPT tokenizers are open source on `tiktoken`, they are not released in a format compatible with HuggingFace tokenizers. We used the `tokenizers`-compatible files uploaded by a HuggingFace user named Xenova. For instance, the GPT-4O tokenizer can be found at `https://huggingface.co/Xenova/gpt-4o`.

### C.6 Discussion of `sentencepiece` tokenizers

The LLAMA and GEMMA tokenizers are trained with the `sentencepiece` library, which uses the same BPE algorithm, except the units of the base vocabulary are characters rather than bytes. In other words, the merge rules learned will apply to pairs of character sequences instead of byte sequences. While byte-level tokenizers always start with the same base vocabulary of 256 bytes, character-level tokenizers determine the base vocabulary using a *character coverage* hyperparameter (usually set to ~0.9995), which determines what proportion of characters that appear in the training text will be in the base vocabulary. Byte fallback is used to represent the out-of-vocabulary characters using bytes.

To empirically test our attack on character-level BPE tokenizers, we train 100 `sentencepiece` tokenizers on mixtures of $n = 10$ natural languages. We apply our attack using the top $T = 3000$ merges, but otherwise use the same settings as §4.

With character-level tokenizers, we achieve an average log MSE of $-4.09$ compared to $-1.39$ for random guessing and $-7.65$ for byte-level tokenizers. That is, character-level tokenizers are harder for our algorithm to reverse than byte-level tokenizers! We believe this is because different languages have widely varying numbers of characters in their writing systems. For languages with many characters, their merges will appear lower in the merge list due to their lower average frequency compared to languages with fewer characters. This leads to a bias in representation among the first $T$ merges considered by our approach.

### C.7 Miscellaneous observation: tie-breaking in `sentencepiece` tokenizers

We observe that in `sentencepiece` tokenizers, after going deep in the merge list (about halfway), the merges begin forming groups, in which the length of the merge is ordered from shortest to longest. We trace this to how ties in pair counts are broken in `sentencepiece`, which is by length of the merge. In terms of reverse engineering, this points to a way to infer the size of the tokenizer training data, since exact ties in frequency become less likely as more training data is considered. We leave this direction to future work.

Table 2: **Top 50 merges of commercial tokenizers we study.** For LLAMA 3*, because the first 100K merges are borrowed directly from GPT-3.5, we show the top 50 merges after that. For readability, we replace the space token Ġ with _ and the newline Ċ with \n. To a careful observer, there are many interpretable signatures of training sources. For instance, consecutive whitespace is common in code, as indents are equivalent to four spaces by most coding standards. Indeed, _ _ is the first merge of all tokenizers that consider merges of whitespace *except* GPT-2 (see §C.4 for relevant discussion). The merges ; \n, _ =, and sel f are also common token pairs in code. The odd-looking symbols are encodings of bytes that make up *parts* of single characters in many languages. For instance, à ¤ is the prefix for characters in the first half of the Devanagari Unicode block (used for writing Hindi, Nepali, Sanskrit, among others), Ð ° encodes the Cyrillic "a" (used in e.g., Russian), and á ĥ encodes the prefix for the second half of the Georgian Unicode block.

| GPT-2 | GPT-3.5 | GPT-4o | LLAMA | LLAMA 3* |
|---|---|---|---|---|
| _ t | _ _ | _ _ | _ t | _ Ù |
| _ a | -- -- | -- -- | e r | Ø§ Ù |
| h e | i n | i n | i n | à¸² à¸ |
| i n | _ t | e r | _ a | Ñ Ł |
| r e | ---- ---- | _ t | e n | ÑŁ ÑŁ |
| o n | e r | _ a | o n | _ à¸ |
| _t he | -- _ | e n | _t h | à¹G à¸ |
| e r | o n | o n | e s | i á» |
| _ s | _ a | r e | _ s | ãGÇãGÇ ãGÇãGÇ |
| a t | r e | _ s | _ d | _Ø§ Ø |
| _ w | a t | a t | a t | à¥ Ī |
| _ o | s t | o r | o r | _ ãGÇ |
| e n | e n | e s | a n | Ñ Ĺ |
| _ c | o r | ---- ---- | _ c | i á»ĩ |
| i t | _t h | a n | i s | ÑŁÑŁ ÑŁÑŁ |
| i s | \n \n | -- _ | r e | à¥ĩà¤ Ĥ |
| a n | _ c | _ d | i t | Ñĸ Ð´ |
| o r | l e | h e | _th e | à¤¾à¤ ° |
| e s | _ s | _ c | a r | ÙĨ Ø |
| _ b | i t | _ p | l e | Ñĸ Ð² |
| e d | a n | i s | _ w | _à¤ ¬ |
| _ f | a r | a r | _ p | _à¤ ľ |
| in g | a l | i t | o u | à¥ ¤ |
| _ p | _t he | \n \n | a l | Ð½ Ñĸ |
| o u | ; \n | a l | _ f | à¤ Ĺ |
| _a n | _ p | à ¤ | _ m | _Ø ¢ |
| a l | _ f | l e | e d | _à¤ ¨ |
| a r | o u | o u | _ o | Ñ Ķ |
| _t o | _ = | _ m | _ b | _ÑĠ Ð° |
| _ m | i s | _ f | o m | _à¤ h |
| _o f | ---- --- | _ w | i on | Ñġ ÑĮ |
| _ in | in g | _ b | in g | _à¤ µ |
| _ d | e s | a s | i c | ÑĪ Ñĸ |
| _ h | _ w | in g | a s | _v á» |
| _an d | i on | _t he | e l | ³ Øª |
| i c | e d | i c | en t | _à¤ ¦ |
| a s | i c | e t | _ in | n ÄŁ |
| l e | _ b | _ o | _ h | _à¤ ² |
| _t h | _ d | i on | n d | _ãGÇ _ãGÇ |
| i on | e t | e d | e t | à¥ Ĥ |
| o m | _ m | e l | _ l | à¤ ¦ |
| l l | _ o | _ n | _ n | à¸Ń à¸ĩ |
| en t | ĉ ĉ | r o | s t | ÙĪ ÙĬ |
| _ n | r o | en t | _t o | à¤ µ |
| _ l | a s | _ Ð | c h | a ÅŁ |
| s t | e l | n d | _ I | à¹ Ĥ |
| _ re | c t | s t | r o | Î¹ Îº |
| v e | n d | á ĥ | i l | _à¤ ° |
| _ e | _ in | Ð ° | _o f | _Ð² Ð¸ |
| r o | _ h | _ l | d e | à¥¡à¤ ¯ |

Table 3: **Top 50 merges of commercial tokenizers we study,** *continued from previous page*.

| MISTRAL | MISTRAL NEMO | GPT-NEOX | CLAUDE | GEMMA |
|---|---|---|---|---|
| _ t | _ _ | _ _ | _ _ | i n |
| i n | _ t | _ t | __ __ | _ t |
| e r | e r | _ a | i n | e r |
| _ a | i n | h e | __ _ | _ a |
| h e | __ __ | i n | _ t | o n |
| o n | _ a | r e | e r | r e |
| r e | e n | o n | ____ ____ | e n |
| _ s | o n | __ __ | o n | h e |
| e n | e s | _t he | _ a | a n |
| a t | _ s | e r | r e | a t |
| o r | _ d | a t | a t | o r |
| _t he | \n \n | _ s | s e | e s |
| e s | h e | e n | h e | _ s |
| _ w | a t | _ o | o r | a r |
| a n | o r | _ w | s t | t i |
| _ c | a n | _ c | e n | t e |
| i s | _ c | i s | ____ ___ | t h |
| i t | r e | i t | a l | s t |
| o u | _ p | o r | _t he | n d |
| _ d | i s | e d | i t | a l |
| a l | i t | e s | _ c | _ o |
| a r | a r | a n | a n | l e |
| _ p | _t he | a l | l e | d e |
| _ f | a l | _ p | _ = | _ i |
| e d | Ø § | _ f | d e | s e |
| _ b | _ o | _ b | a r | _ c |
| in g | l e | _a n | \n _______ | _ d |
| _ o | _ m | in g | _ f | i t |
| _ m | _ f | _o f | _ p | n t |
| l e | _ w | a r | \n ________ | i s |
| n d | e d | _ in | _ o | _ p |
| a s | â Ģ | o u | _ s | m e |
| i c | a s | _ d | _ w | r i |
| _ h | _ b | _ m | m e | r a |
| i on | i c | i on | \n ___ | o u |
| _ in | r o | i c | r o | a s |
| _t o | i on | _t o | i on | e d |
| e t | __ _ | l e | in g | n e |
| o m | _ in | - - | i s | t o |
| e l | _ l | a s | _ in | n g |
| _o f | ã Ģ | _an d | _ b | _ w |
| s t | en t | ____ ____ | i c | r o |
| _a nd | n d | r o | se l | l i |
| _ l | e l | _ h | o u | t a |
| _t h | _ Đ | _t h | sel f | _ f |
| _ n | ____ ____ | en t | e d | _ b |
| en t | in g | c t | - - | _ m |
| i l | Ũ Ħ | e t | n d | i c |
| c t | e t | e l | e s | e l |
| r o | o u | _ re | _ m | l a |

Table 4: **Our full set of inferences for commercial tokenizers** over 116 categories (111 languages, 4 English domains, and code). The four English domains are web, books, academic, and Wikipedia.

| Category | GPT-2 | GPT-3.5 | GPT-4o | LLAMA | LLAMA 3* | MISTRAL | MISTRAL NeMo | GPT-NeoX | CLAUDE | GEMMA |
|---|---|---|---|---|---|---|---|---|---|---|
| Web | 83.6 | 27.3 | 20.7 | 11.3 | 12.7 | 30.3 | 17.1 | 43.7 | 12.7 | 25.7 |
| Code | 0.7 | 62.6 | 32.8 | 19.2 | 30.2 | 25.8 | 18.3 | 15.2 | 57.5 | 25.9 |
| Books | 15.4 | 6.8 | 7.4 | 23.1 | 8.5 | 21.5 | 9.6 | 26.3 | 17.4 | 12.8 |
| Academic | 0.1 | 0.2 | 0.0 | 8.0 | 0.1 | 5.3 | 3.7 | 12.0 | 5.1 | 4.3 |
| Wiki | 0.0 | 0.0 | 0.0 | 6.7 | 0.0 | 0.5 | 4.6 | 0.0 | 3.7 | 3.0 |
| French | 0.0 | 0.3 | 2.9 | 5.3 | 1.8 | 2.3 | 6.3 | 0.2 | 0.0 | 3.0 |
| German | 0.0 | 0.4 | 1.8 | 5.1 | 2.2 | 2.1 | 3.2 | 0.1 | 0.1 | 2.8 |
| Arabic | 0.0 | 0.0 | 1.6 | 0.0 | 1.6 | 0.0 | 4.8 | 0.0 | 0.0 | 0.3 |
| Japanese | 0.1 | 0.0 | 0.4 | 0.3 | 4.1 | 0.2 | 1.9 | 0.2 | 0.0 | 1.6 |
| Spanish | 0.0 | 0.6 | 2.8 | 2.7 | 2.0 | 1.7 | 3.7 | 0.1 | 0.0 | 3.9 |
| Russian | 0.0 | 0.1 | 2.8 | 2.6 | 3.4 | 1.2 | 2.6 | 0.2 | 0.1 | 1.4 |
| Turkish | 0.0 | 0.0 | 0.6 | 0.0 | 3.2 | 0.1 | 0.4 | 0.0 | 0.0 | 0.6 |
| Czech | 0.0 | 0.0 | 0.3 | 0.6 | 2.7 | 0.4 | 0.5 | 0.0 | 0.0 | 0.3 |
| Ukrainian | 0.0 | 0.0 | 0.3 | 1.3 | 2.7 | 0.6 | 0.6 | 0.0 | 0.0 | 0.2 |
| Italian | 0.0 | 0.3 | 0.5 | 2.6 | 1.0 | 1.5 | 1.7 | 0.1 | 0.3 | 1.6 |
| Korean | 0.0 | 0.0 | 0.7 | 0.1 | 2.6 | 0.1 | 2.5 | 0.0 | 0.1 | 0.2 |
| Portuguese | 0.0 | 0.3 | 2.3 | 1.1 | 1.4 | 0.5 | 1.5 | 0.3 | 0.5 | 1.5 |
| Persian | 0.0 | 0.0 | 0.4 | 0.0 | 2.2 | 0.0 | 1.2 | 0.0 | 0.0 | 0.3 |
| Dutch | 0.0 | 0.1 | 2.0 | 1.2 | 0.9 | 0.6 | 0.6 | 0.1 | 0.1 | 0.6 |
| Vietnamese | 0.0 | 0.0 | 0.5 | 0.1 | 1.8 | 0.0 | 0.6 | 0.0 | 0.0 | 0.4 |
| Greek | 0.0 | 0.0 | 0.6 | 0.0 | 1.7 | 0.0 | 0.7 | 0.1 | 0.0 | 0.1 |
| Thai | 0.0 | 0.0 | 0.4 | 0.0 | 1.7 | 0.0 | 0.2 | 0.0 | 0.0 | 0.1 |
| Hindi | 0.0 | 0.0 | 1.4 | 0.0 | 1.7 | 0.0 | 1.0 | 0.0 | 0.0 | 0.1 |
| Polish | 0.0 | 0.1 | 0.5 | 1.4 | 1.5 | 0.7 | 0.8 | 0.1 | 0.0 | 0.7 |
| Catalan | 0.0 | 0.0 | 0.3 | 1.1 | 0.3 | 0.4 | 0.9 | 0.0 | 0.3 | 0.4 |
| Georgian | 0.0 | 0.0 | 0.8 | 0.0 | 0.0 | 0.0 | 0.3 | 0.0 | 0.0 | 0.0 |
| Indonesian | 0.0 | 0.1 | 0.4 | 0.1 | 0.3 | 0.0 | 0.4 | 0.0 | 0.0 | 0.8 |
| Chinese | 0.0 | 0.0 | 0.8 | 0.0 | 0.3 | 0.0 | 0.2 | 0.0 | 0.1 | 0.1 |
| Swedish | 0.0 | 0.0 | 0.3 | 0.7 | 0.5 | 0.5 | 0.2 | 0.1 | 0.0 | 0.1 |
| Estonian | 0.0 | 0.1 | 0.5 | 0.1 | 0.5 | 0.1 | 0.1 | 0.1 | 0.2 | 0.6 |
| Finnish | 0.0 | 0.0 | 0.5 | 0.2 | 0.6 | 0.1 | 0.3 | 0.1 | 0.0 | 0.6 |
| Low German | 0.0 | 0.0 | 0.0 | 0.6 | 0.1 | 0.1 | 0.4 | 0.0 | 0.0 | 0.0 |
| Serbian | 0.0 | 0.0 | 0.1 | 0.5 | 0.0 | 0.2 | 0.6 | 0.0 | 0.0 | 0.0 |
| Gujarati | 0.0 | 0.0 | 0.6 | 0.0 | 0.0 | 0.0 | 0.1 | 0.0 | 0.0 | 0.0 |
| Malayalam | 0.0 | 0.0 | 0.6 | 0.0 | 0.0 | 0.0 | 0.2 | 0.0 | 0.0 | 0.0 |
| Bangla | 0.0 | 0.0 | 0.5 | 0.0 | 0.0 | 0.0 | 0.5 | 0.0 | 0.0 | 0.0 |
| Galician | 0.0 | 0.0 | 0.2 | 0.3 | 0.1 | 0.0 | 0.5 | 0.1 | 0.2 | 0.0 |
| Hebrew | 0.0 | 0.0 | 0.5 | 0.0 | 0.0 | 0.0 | 0.4 | 0.0 | 0.0 | 0.1 |
| Armenian | 0.0 | 0.0 | 0.5 | 0.0 | 0.0 | 0.0 | 0.5 | 0.0 | 0.0 | 0.0 |
| Basque | 0.0 | 0.0 | 0.1 | 0.0 | 0.3 | 0.1 | 0.2 | 0.0 | 0.1 | 0.5 |
| Telugu | 0.0 | 0.0 | 0.4 | 0.0 | 0.0 | 0.0 | 0.5 | 0.0 | 0.0 | 0.0 |
| Romanian | 0.0 | 0.0 | 0.3 | 0.4 | 0.5 | 0.2 | 0.4 | 0.1 | 0.1 | 0.3 |
| Filipino | 0.0 | 0.0 | 0.3 | 0.1 | 0.2 | 0.0 | 0.0 | 0.0 | 0.1 | 0.5 |
| Lithuanian | 0.0 | 0.0 | 0.1 | 0.0 | 0.1 | 0.1 | 0.1 | 0.0 | 0.0 | 0.5 |
| Danish | 0.0 | 0.0 | 0.4 | 0.3 | 0.5 | 0.4 | 0.1 | 0.1 | 0.3 | 0.2 |
| Bulgarian | 0.0 | 0.0 | 0.2 | 0.2 | 0.2 | 0.2 | 0.4 | 0.0 | 0.0 | 0.1 |
| Kannada | 0.0 | 0.0 | 0.4 | 0.0 | 0.0 | 0.0 | 0.3 | 0.0 | 0.0 | 0.0 |
| Welsh | 0.0 | 0.0 | 0.1 | 0.1 | 0.2 | 0.1 | 0.1 | 0.1 | 0.0 | 0.4 |
| Slovenian | 0.0 | 0.1 | 0.3 | 0.2 | 0.3 | 0.1 | 0.3 | 0.0 | 0.2 | 0.1 |
| Urdu | 0.0 | 0.0 | 0.3 | 0.0 | 0.0 | 0.0 | 0.2 | 0.0 | 0.0 | 0.0 |
| Malagasy | 0.0 | 0.0 | 0.1 | 0.1 | 0.1 | 0.0 | 0.0 | 0.0 | 0.1 | 0.3 |
| Tamil | 0.0 | 0.0 | 0.3 | 0.0 | 0.0 | 0.0 | 0.3 | 0.0 | 0.0 | 0.0 |
| Irish | 0.0 | 0.0 | 0.3 | 0.1 | 0.1 | 0.1 | 0.1 | 0.0 | 0.0 | 0.2 |
| Tajik | 0.0 | 0.0 | 0.3 | 0.0 | 0.0 | 0.0 | 0.0 | 0.0 | 0.0 | 0.0 |
| Macedonian | 0.0 | 0.0 | 0.1 | 0.1 | 0.1 | 0.1 | 0.3 | 0.0 | 0.0 | 0.0 |
| Nepali | 0.0 | 0.0 | 0.3 | 0.0 | 0.0 | 0.0 | 0.0 | 0.0 | 0.0 | 0.0 |
| Kazakh | 0.0 | 0.0 | 0.3 | 0.0 | 0.0 | 0.0 | 0.1 | 0.0 | 0.0 | 0.0 |
| Belarusian | 0.0 | 0.0 | 0.3 | 0.1 | 0.1 | 0.1 | 0.3 | 0.0 | 0.0 | 0.0 |
| Esperanto | 0.0 | 0.0 | 0.0 | 0.2 | 0.1 | 0.1 | 0.3 | 0.0 | 0.0 | 0.1 |
| Afrikaans | 0.0 | 0.0 | 0.3 | 0.2 | 0.1 | 0.2 | 0.2 | 0.0 | 0.1 | 0.2 |
| Tatar | 0.0 | 0.0 | 0.3 | 0.0 | 0.0 | 0.0 | 0.0 | 0.0 | 0.0 | 0.0 |
| Norwegian | 0.0 | 0.0 | 0.2 | 0.0 | 0.3 | 0.0 | 0.0 | 0.0 | 0.0 | 0.1 |
| Uzbek | 0.0 | 0.0 | 0.1 | 0.1 | 0.1 | 0.0 | 0.1 | 0.0 | 0.1 | 0.2 |
| Norwegian Nynorsk | 0.0 | 0.1 | 0.1 | 0.2 | 0.0 | 0.2 | 0.2 | 0.0 | 0.0 | 0.1 |
| Slovak | 0.0 | 0.0 | 0.2 | 0.0 | 0.2 | 0.0 | 0.1 | 0.0 | 0.0 | 0.2 |
| Lojban | 0.1 | 0.0 | 0.2 | 0.0 | 0.2 | 0.1 | 0.1 | 0.0 | 0.0 | 0.0 |
| Icelandic | 0.0 | 0.0 | 0.2 | 0.1 | 0.1 | 0.0 | 0.0 | 0.0 | 0.0 | 0.1 |
| Kyrgyz | 0.0 | 0.0 | 0.2 | 0.0 | 0.0 | 0.0 | 0.0 | 0.0 | 0.0 | 0.0 |
| Pashto | 0.0 | 0.0 | 0.2 | 0.0 | 0.1 | 0.0 | 0.0 | 0.0 | 0.0 | 0.0 |
| Azerbaijani | 0.0 | 0.0 | 0.1 | 0.0 | 0.1 | 0.0 | 0.2 | 0.0 | 0.0 | 0.1 |
| Breton | 0.0 | 0.0 | 0.1 | 0.1 | 0.1 | 0.1 | 0.1 | 0.0 | 0.0 | 0.2 |
| Yiddish | 0.0 | 0.0 | 0.2 | 0.0 | 0.0 | 0.0 | 0.0 | 0.0 | 0.0 | 0.0 |
| Bashkir | 0.0 | 0.0 | 0.2 | 0.0 | 0.0 | 0.0 | 0.0 | 0.0 | 0.0 | 0.0 |
| Kurdish | 0.0 | 0.0 | 0.1 | 0.0 | 0.2 | 0.0 | 0.0 | 0.0 | 0.0 | 0.1 |
| Hungarian | 0.0 | 0.0 | 0.0 | 0.1 | 0.0 | 0.2 | 0.1 | 0.0 | 0.0 | 0.0 |
| Latvian | 0.0 | 0.0 | 0.1 | 0.0 | 0.1 | 0.0 | 0.1 | 0.0 | 0.0 | 0.2 |
| Albanian | 0.0 | 0.0 | 0.1 | 0.0 | 0.1 | 0.0 | 0.1 | 0.0 | 0.0 | 0.1 |
| Assamese | 0.0 | 0.0 | 0.1 | 0.0 | 0.0 | 0.0 | 0.0 | 0.0 | 0.0 | 0.0 |
| Luxembourgish | 0.0 | 0.0 | 0.0 | 0.1 | 0.1 | 0.0 | 0.0 | 0.0 | 0.0 | 0.1 |
| Cebuano | 0.0 | 0.1 | 0.1 | 0.0 | 0.1 | 0.0 | 0.0 | 0.0 | 0.1 | 0.0 |
| Amharic | 0.0 | 0.0 | 0.0 | 0.0 | 0.0 | 0.0 | 0.0 | 0.0 | 0.0 | 0.1 |
| Sindhi | 0.0 | 0.0 | 0.1 | 0.0 | 0.0 | 0.0 | 0.1 | 0.0 | 0.0 | 0.0 |
| Ossetic | 0.0 | 0.0 | 0.1 | 0.0 | 0.0 | 0.0 | 0.0 | 0.0 | 0.0 | 0.0 |
| Western Frisian | 0.0 | 0.0 | 0.0 | 0.0 | 0.0 | 0.0 | 0.0 | 0.0 | 0.0 | 0.1 |

*Table continues...*

| Category | GPT-2 | GPT-3.5 | GPT-4o | LLAMA | LLAMA 3* | MISTRAL | MISTRAL NEMO | GPT-NEOX | CLAUDE | GEMMA |
|---|---|---|---|---|---|---|---|---|---|---|
| Chechen | 0.0 | 0.0 | 0.1 | 0.1 | 0.1 | 0.0 | 0.0 | 0.0 | 0.0 | 0.0 |
| Piedmontese | 0.0 | 0.0 | 0.1 | 0.1 | 0.0 | 0.0 | 0.1 | 0.0 | 0.0 | 0.0 |
| Turkmen | 0.0 | 0.0 | 0.1 | 0.0 | 0.1 | 0.0 | 0.0 | 0.0 | 0.0 | 0.1 |
| Mongolian | 0.0 | 0.0 | 0.1 | 0.0 | 0.0 | 0.0 | 0.1 | 0.0 | 0.0 | 0.0 |
| Burmese | 0.0 | 0.0 | 0.1 | 0.0 | 0.0 | 0.0 | 0.1 | 0.0 | 0.0 | 0.0 |
| South Azerbaijani | 0.0 | 0.0 | 0.1 | 0.0 | 0.1 | 0.0 | 0.0 | 0.0 | 0.0 | 0.0 |
| Marathi | 0.0 | 0.0 | 0.1 | 0.0 | 0.0 | 0.0 | 0.1 | 0.0 | 0.0 | 0.0 |
| Latin | 0.0 | 0.0 | 0.1 | 0.0 | 0.0 | 0.0 | 0.0 | 0.0 | 0.1 | 0.1 |
| Uyghur | 0.0 | 0.0 | 0.1 | 0.0 | 0.0 | 0.0 | 0.1 | 0.0 | 0.0 | 0.0 |
| Sinhala | 0.0 | 0.0 | 0.1 | 0.0 | 0.0 | 0.0 | 0.0 | 0.0 | 0.0 | 0.0 |
| Punjabi | 0.0 | 0.0 | 0.1 | 0.0 | 0.0 | 0.0 | 0.1 | 0.0 | 0.0 | 0.0 |
| Chuvash | 0.0 | 0.0 | 0.0 | 0.1 | 0.0 | 0.0 | 0.0 | 0.0 | 0.0 | 0.0 |
| Khmer | 0.0 | 0.0 | 0.1 | 0.0 | 0.0 | 0.0 | 0.0 | 0.0 | 0.0 | 0.0 |
| Western Panjabi | 0.0 | 0.0 | 0.0 | 0.0 | 0.0 | 0.0 | 0.1 | 0.0 | 0.0 | 0.0 |
| Maltese | 0.0 | 0.0 | 0.0 | 0.0 | 0.0 | 0.0 | 0.0 | 0.0 | 0.0 | 0.0 |
| Newari | 0.0 | 0.0 | 0.0 | 0.0 | 0.0 | 0.0 | 0.0 | 0.0 | 0.0 | 0.0 |
| Lao | 0.0 | 0.0 | 0.0 | 0.0 | 0.0 | 0.0 | 0.0 | 0.0 | 0.0 | 0.0 |
| Sakha | 0.0 | 0.0 | 0.0 | 0.0 | 0.0 | 0.0 | 0.0 | 0.0 | 0.0 | 0.0 |
| Croatian | 0.0 | 0.0 | 0.0 | 0.0 | 0.0 | 0.0 | 0.0 | 0.0 | 0.0 | 0.0 |
| Mingrelian | 0.0 | 0.0 | 0.0 | 0.0 | 0.0 | 0.0 | 0.0 | 0.0 | 0.0 | 0.0 |
| Sanskrit | 0.0 | 0.0 | 0.0 | 0.0 | 0.0 | 0.0 | 0.0 | 0.0 | 0.0 | 0.0 |
| Central Kurdish | 0.0 | 0.0 | 0.0 | 0.0 | 0.0 | 0.0 | 0.0 | 0.0 | 0.0 | 0.0 |
| Eastern Mari | 0.0 | 0.0 | 0.0 | 0.0 | 0.0 | 0.0 | 0.0 | 0.0 | 0.0 | 0.0 |
| Swahili | 0.0 | 0.0 | 0.0 | 0.0 | 0.0 | 0.0 | 0.0 | 0.0 | 0.0 | 0.0 |
| Odia | 0.0 | 0.0 | 0.0 | 0.0 | 0.0 | 0.0 | 0.0 | 0.0 | 0.0 | 0.0 |
| Bishnupriya | 0.0 | 0.0 | 0.0 | 0.0 | 0.0 | 0.0 | 0.0 | 0.0 | 0.0 | 0.0 |
| Egyptian Arabic | 0.0 | 0.0 | 0.0 | 0.0 | 0.0 | 0.0 | 0.0 | 0.0 | 0.0 | 0.0 |
| Tibetan | 0.0 | 0.0 | 0.0 | 0.0 | 0.0 | 0.0 | 0.0 | 0.0 | 0.0 | 0.0 |
| Divehi | 0.0 | 0.0 | 0.0 | 0.0 | 0.0 | 0.0 | 0.0 | 0.0 | 0.0 | 0.0 |
| Minangkabau | 0.0 | 0.0 | 0.0 | 0.0 | 0.0 | 0.0 | 0.0 | 0.0 | 0.0 | 0.0 |
| Goan Konkani | 0.0 | 0.0 | 0.0 | 0.0 | 0.0 | 0.0 | 0.0 | 0.0 | 0.0 | 0.0 |
| Malay | 0.0 | 0.0 | 0.0 | 0.0 | 0.0 | 0.0 | 0.0 | 0.0 | 0.0 | 0.0 |

Table 5: The 112 natural languages considered in §4. The data is from Oscar v23.01, which performs language identification at the document level.

| Language | Size (MB) | Language | Size (MB) | Language | Size (MB) | Language | Size (MB) |
|---|---|---|---|---|---|---|---|
| Chinese | 776494.9 | Bangla | 19055.4 | Icelandic | 2194.7 | Sanskrit | 56.3 |
| English | 666955.4 | Hebrew | 17970.6 | Slovenian | 1398.1 | Ossetic | 50.7 |
| Russian | 531902.4 | Tamil | 15776.8 | Punjabi | 1377.2 | Chuvash | 42.3 |
| Spanish | 424143.2 | Catalan | 15346.5 | Basque | 1195.9 | Cebuano | 41.1 |
| French | 371967.1 | Danish | 14843.6 | Tajik | 1028.4 | Afrikaans | 37.2 |
| German | 356683.7 | Lithuanian | 14518.6 | Tatar | 834.1 | Breton | 31.4 |
| Italian | 214768.2 | Georgian | 8388.2 | Central Kurdish | 773.1 | South Azerbaijani | 28.4 |
| Japanese | 181299.8 | Estonian | 8026.9 | Filipino | 719.4 | Croatian | 26.5 |
| Hungarian | 150134.4 | Serbian | 7666.2 | Odia | 543.2 | Eastern Mari | 22.9 |
| Polish | 146001.9 | Latvian | 7411.5 | Tibetan | 531.6 | Luxembourgish | 18.4 |
| Vietnamese | 139298.4 | Malayalam | 5815.1 | Amharic | 513.0 | Uzbek | 15.3 |
| Dutch | 135078.1 | Mongolian | 5777.3 | Kyrgyz | 489.5 | Chechen | 13.9 |
| Arabic | 110728.5 | Gujarati | 5593.9 | Esperanto | 475.1 | Malagasy | 11.2 |
| Portuguese | 105065.0 | Nepali | 4950.5 | Lao | 472.3 | Low German | 10.7 |
| Greek | 95750.9 | Armenian | 4884.7 | Assamese | 412.2 | Mingrelian | 6.1 |
| Persian | 93225.0 | Macedonian | 4745.4 | Bashkir | 363.9 | Bishnupriya | 5.4 |
| Thai | 91968.7 | Marathi | 4478.3 | Welsh | 333.1 | Newari | 4.0 |
| Czech | 76987.1 | Telugu | 3873.8 | Pashto | 261.7 | Minangkabau | 3.8 |
| Turkish | 72207.2 | Urdu | 3761.3 | Galician | 255.9 | Egyptian Arabic | 3.7 |
| Swedish | 50001.1 | Kazakh | 3325.4 | Uyghur | 219.8 | Norwegian Nynorsk | 3.7 |
| Romanian | 45590.6 | Albanian | 3224.9 | Divehi | 200.2 | Turkmen | 3.3 |
| Ukrainian | 44746.7 | Khmer | 3155.2 | Kurdish | 174.2 | Piedmontese | 3.1 |
| Bulgarian | 44118.5 | Azerbaijani | 3038.3 | Yiddish | 171.8 | Malay | 2.6 |
| Finnish | 41143.7 | Burmese | 3035.4 | Sindhi | 131.7 | Goan Konkani | 2.3 |
| Korean | 38158.4 | Sinhala | 2599.3 | Western Panjabi | 105.8 | Latin | 2.0 |
| Hindi | 32615.5 | Norwegian | 2583.2 | Western Frisian | 70.5 | Lojban | 1.5 |
| Indonesian | 23416.0 | Kannada | 2574.4 | Sakha | 68.8 | Maltese | 1.3 |
| Slovak | 21460.7 | Belarusian | 2339.5 | Irish | 63.2 | Swahili | 1.0 |

Table 6: The 37 programming languages considered in §4. Data is sourced from the Github split of RedPajama, and classified into a language based on the file extension.

| Language | Size (in MB) | Language | Size (in MB) |
|---|---|---|---|
| Java | 29493.0 | Haskell | 547.6 |
| JavaScript | 27910.4 | TSQL | 489.5 |
| HTML | 25864.1 | Lua | 393.4 |
| XML | 18804.0 | Dockerfile | 272.7 |
| C++ | 15543.1 | Makefile | 265.7 |
| Python | 12970.1 | TeX | 256.9 |
| Smalltalk | 11580.5 | XPixMap | 248.7 |
| Objective-C | 10909.5 | PowerShell | 240.7 |
| PHP | 9837.4 | CMake | 118.5 |
| Go | 6287.2 | Raku | 106.9 |
| Markdown | 6137.3 | Hack | 79.1 |
| C | 6045.0 | Julia | 72.3 |
| CSS | 4084.9 | Batchfile | 60.9 |
| Ruby | 3381.4 | Pod6 | 46.6 |
| Scala | 1376.8 | FortranFreeForm | 40.8 |
| Smali | 978.3 | Fortran | 31.2 |
| reStructuredText | 891.4 | Motorola68KAssembly | 22.7 |
| VisualBasic.NET | 563.0 | Perl | 2.0 |
| Shell | 551.6 | | |

Table 7: The 5 domains considered in §4. Data is sourced from RedPajama.

| Domain | Size (in MB) |
|---|---|
| Web | 305139.9 |
| Code | 196506.0 |
| Books | 104975.0 |
| Academic | 89044.9 |
| Wikipedia | 20505.8 |

