# OpenReview forum: "Data Mixture Inference Attack: BPE Tokenizers Reveal Training Data Compositions"
_NeurIPS.cc/2024/Conference — NeurIPS 2024 poster_

### Official Review · Reviewer_vDkN · 2024-07-10

**Soundness:** 2
**Presentation:** 3
**Contribution:** 2
**Rating:** 7
**Confidence:** 4

**Summary:**

This paper introduces the task of  data mixture inference, i.e. trying to infer what kind of data (e.g. languages, code) a given LLM is trained on. They do so by leveraging the ordered merge rules learned by the LLM's BPE tokenizer. First, they formalize the problem. Next, authors propose a neatly explained solution using linear programming, with multiple computational optimizations which makes the inference feasible in practice. They then apply their method to a controlled setup, against models for which the data mixture is known and show that their method reaches a significantly better performance than random guessing. Lastly, they apply their method to widely used LLMs for which the mixture is not known and provide their inference results.

**Strengths:**

- Originality: The paper proposes the novel task of data mixture inference, and provides a compelling method to do so.
- Quality: Both the task and proposed method is presented in a compelling way. The task of data mixture inference is formalized, and a novel, technically developed method is proposed. Both the evaluation in a controlled setup as well as the application of the method on other models for which the data mixture is unknown is convincing and interesting. From an experimental perspective, authors should provide a more sophisticated baseline than random guessing. Only then will the superiority of the proposed method become clear. Also, authors could elaborate more on potential defenses against the attack (see questions).
- Clarity: the paper is written very clearly.
- Significance: The paper proposes an attack able to discover a fundamental design decision made by model developers, which they likely want to keep proprietary. Exposing this attack, with its results is quite valuable to the field.

**Weaknesses:**

- Originality: NA
- Quality: From an experimental perspective, authors should provide a more sophisticated baseline than random guessing. Only then will the superiority of the proposed method become clear. Also, authors could elaborate more on potential defenses against the attack (see questions).
- Clarity: NA
- Significance: A more elaborate discussion on potential defenses would increase the significance of the work's contribution.

**Questions:**

- When I think about the relevance of the findings represented in the paper, I believe that this mostly exposes a vulnerability to discover fundamental design decisions used by model developers, which they likely want to keep proprietary. From that perspective, could authors elaborate more on potential defenses and how they could perform? For instance, I do not fully understand why post-hoc re-ordering of the merge rules would hurt the utility of the tokenizer (maybe you can show this?). And if this is not a viable option, is the only defense then to not reveal any details on the tokenizer?
- While the task at hand is novel, the only baseline considered (random guessing) is rather naïve. Could you include results of more elaborate baselines (e.g. leveraging the encoding efficiency of the tokenizer for each data sample? Or for the open source models, leveraging the perplexity of the model computed on the data samples)? This would shed more light into how difficult this problem might be and how valuable the proposed linear programming attack and its computational optimizations truly are.
- One assumption that is made throughout the paper is that the attacker knows all potential data sources to include. Could you add an experiment on how the MSE changes when not all data sources are known (eg not including X natural languages, and have an increasing fraction of 'other data')?

**Limitations:**

Authors should elaborate more on the potential defenses against this attack.

---

> ### Author Rebuttal · Authors · 2024-08-07
>
> Thank you for recognizing the “novel” task we study, our “technically developed method,” and the “convincing,” "valuable results" on a "fundamental design decision" made by model developers.
>
> Thank you for the thoughtful questions, and we hope that our response addresses your concerns!
>
> ## Elaboration on defense based on merge list reordering
> Consider a merge list with three entries,
>
> (`h`, `e`)
>
> (`t`, `h`)
>
> (`t`, `he`)
>
> Observe that to tokenize the word “the,” first (`h`, `e`) is merged, then (`t`, `he`). Thus, “the” is encoded as a single token.
>
> Now suppose the first two merges are flipped. Then the merge (`t`, `h`) would be applied first, and (`t`, `he`) is no longer applicable. Now, “the” would be represented as a sequence of two tokens, [`th`, `e`]. This is a problem because “the” is represented in a completely different way than in training!
>
> However if we instead consider a new merge, (`i`, `n`), we see that it can be placed anywhere in the current merge list. This is because it is completely independent of the current merges, as they do not share any tokens.
>
> Thus, *some* functionality-preserving reorderings are possible. However, because these are fairly limited, we expect the attack to still be applicable as described in the paper.
>
> ## Our attack is surprisingly robust to unaccounted-for categories
>
> Following your suggestion, we run **new experiments simulating the setting where only a subset of true categories are known**. We find that our attack is surprisingly robust — **even when $1/10$ of the probability mass is unaccounted for, our attack achieves $\log_{10}$ MSE around $-6$**, compared to $-7.69$ with the full set of languages. *Please see the global response for details.*
>
> ## New experiments show our attack beats baseline based on tokenizer encoding efficiency
>
> Thank you for your suggestion to try a baseline based on the tokenizer’s performance on each language. We implement this baseline and find that this baseline achieves $\log_{10}$ MSE of $-2.29$, compared to $-1.84$ random guessing and $-7.66$ for our attack. Thus **while the baseline is meaningfully better than random, our attack is still $10^5 \times$ stronger!** This highlights the difficulty of the problem and the effectiveness of our attack, and we will definitely include this comparison in the next revision. *Please see the global response for details.*

---

> > ### Comment · Reviewer_vDkN · 2024-08-08
> > **Answer to rebuttal**
> >
> > Thank you for your response, and including additional results. I think this is a solid paper, with a novel and interesting problem statement and elaborate and convincing experimental results.
> >
> > I just bumped up my score to a 7. Hope it gets accepted.

---

### Official Review · Reviewer_68Uz · 2024-07-11

**Soundness:** 3
**Presentation:** 3
**Contribution:** 2
**Rating:** 5
**Confidence:** 3

**Summary:**

This paper proposes a novel data mixture inference attack to uncover the distributional makeup of pretraining data for large language models. By leveraging the ordered vocabulary learned by byte-pair encoding (BPE) tokenizers, the authors formulate a linear program to infer the relative proportions of different data categories (e.g., languages, programming languages, data sources) in the tokenizer's training set.

**Strengths:**

1.	The paper introduces a previously understudied problem of data mixture inference, complementing existing membership inference attacks that focus on individual instances.
2.	The proposed attack method, based on BPE tokenizers' ordered merge rules, giving some insightful findings in inferring data mixture proportions.

**Weaknesses:**

1.	The attack relies on the specific behavior and structure of BPE tokenizers, which limits the generalizability of the method to different tokenizers.
2.	The attack hinges on the representativeness of tokenizer training data and the quality of sample data (from the same distribution in the paper), which may not always hold.

**Questions:**

1.	What the meaning of random guessing? Are there alternative methods to infer the proportions of different categories, possibly leveraging model performance metrics or behavioral patterns?
2.	What strategies can be employed to ensure the quality of sample data for estimating pair frequencies.

**Limitations:**

Yes

---

> ### Author Rebuttal · Authors · 2024-08-07
>
> Thank you for recognizing the “previously understudied” problem we address and the "insightful findings." We have taken care to investigate all the questions you raise (with new experiments) and hope that our response addresses your concerns.
>
> ## Today’s LLMs use BPE
> While it is true that other tokenization algorithms may exist, BPE is the universal choice for today’s production models; in fact, we manually verified that **all top 100 models on ChatbotArena use BPE tokenizers**. In the future, powerful LMs that use tokenizers other than BPE may be released, but we cannot guess what algorithm they will use.
>
> ## New experiments show that the attack is robust to distribution shift
> Following your suggestion, we run **new experiments where there is an extreme distribution shift between the training data of the tokenizer and the data available to the attacker**. We show that the attack performance degrades gracefully in this setting, **remaining strong even under extreme shift**. *Please see the global response for more details.*
>
> ## New experiments show our attack beats baseline based on tokenizer encoding efficiency
> Thank you for your insightful suggestion to try a baseline based on the tokenizer’s performance on each language! We implement this baseline and find that it baseline achieves $\log_{10}$ MSE of $-2.29$, compared to $-1.84$ random guessing and $-7.66$ for our attack. *Please see the global response for details.* Thus **while the baseline is meaningfully better than random, our attack is still $10^5 \times$ stronger**! This highlights the difficulty of the problem and the effectiveness of our attack, and we will definitely include this comparison in the next revision.
>
> ## Design considerations for sample data
> As an attacker, one should use any information available to strengthen the attack. If the attacker knows that the training data has a certain property, it would be beneficial to use that kind of data for the pair counting. However it is often the case that the attacker knows very little about the data. In this case, if an attacker has access to multiple data sources, multiple preprocessing techniques etc., it would be beneficial to include all of them as separate categories for the optimization to consider. This increases the chances of finding a mixture that is a good fit for the tokenizer.
>
> Moreover, our new experiments under distribution shift (mentioned above) show empirically that performance remains strong in this setting.
>
> ## Clarification regarding random guessing
> The random guess is calculated by sampling uniformly from the $n$-dimensional simplex, where $n$ is the number of categories. Note that the ground truth values are also sampled this way, so the random guesses are from the “correct” distribution. What we report is the average error over many such guesses.

---

> > ### Comment · Reviewer_68Uz · 2024-08-09
> > **Answer to Rebuttal**
> >
> > Thank you for your response, which addressed my concerns. I have raised the rating from 3 to 5.

---

### Official Review · Reviewer_2xkV · 2024-07-12

**Soundness:** 3
**Presentation:** 3
**Contribution:** 3
**Rating:** 6
**Confidence:** 3

**Summary:**

In this paper, the authors studied inference attack on large language models’ (LLM) tokenizer. In particular, they proposed an attack to inference training data sampling weight used to train tokenizer of LLMs, which usually is also the same sampling weight used to train the LLMs. They formulated their attack as a very large linear programming problem and proposed efficient methods to reduce the size of the linear programming and hence efficiently solve it. Moreover, they evaluated the proposed attack on tokenizers with known training data sampling weights and also some commercial LLMs.

**Strengths:**

The proposed attack seems novel and the problem of interest is very relevant to key problems in LLM area. They verified the effectiveness of their attack on several data mix scenarios. The structure of the paper is also clear and easy to follow.

**Weaknesses:**

I listed a few directions that I think the paper can be improved.
1. Theoretical analysis in this paper is a bit weak. The proposed algorithm in Section 3 does not provide convergence guarantee.  No upper bound provided for the size of the subset of constraints that are violated.  No analysis provided to support that the iterative process will finally provide an $\alpha$ that violate no constraints in LP1.
2. The attack implicitly assumes that one can access the corpus with similar words distribution as the pre-training datasets. However, it is unclear to me how different preprocessing techniques may affect the words distribution of the pre-training dataset and thus the effectiveness of the attack. It will be good to at least exam this impact via numerical experiments.

**Questions:**

1. Pretokenization and the starting vocabulary seems to be very influential on the effectiveness of the proposed algorithm. In this case, how to find a good starting vocabulary/protokenization results?
2. The formulation and Figure 3 assume that the token frequency distribution is proportion to the sampling weight $\alpha_i$, which further assumes that the corpus token frequency is somehow equally distributed across the corpus. Does that make sense? Can we empirically verify these assumptions? Do we need a very large corpus so that these assumptions hold true?
3. How does the vocabulary size, size of the data and number of merges $T$ affect the time the attack requires to run?

---

> ### Author Rebuttal · Authors · 2024-08-07
>
> We thank the reviewer for recognizing the novelty of our attack and its “relevan[ce] to key problems in LLM area.” We have taken all of your suggestions into account and hope that our response addresses your concerns.
>
> ## Theoretical analysis
> Regarding the theoretical analysis, the key observation is that LP2 is a subset of LP1 and its size increases with each iteration. Thus, we are guaranteed to either terminate early or, in the worst case, eventually add all constraints and variables from LP1. Thus, the convergence to a feasible solution is guaranteed. However since LP1 is very large (albeit polynomial in the problem parameters), the hope is that we will terminate early. Unfortunately, proving a strong bound on the number of iterations required would be unprecedented for an algorithm using this approach. (It is typical for papers leveraging row and column generation to prove convergence in finite time and demonstrate that the effective number of iterations is low using experiments.)
>
> We will add a paragraph explaining this reasoning to the paper. Additionally, we notice in practice that the number of iterations required is highly problem dependent, with some instances being solved extremely quickly. We report the worst case timings observed in Section 3.
>
> ## New experiments show that the attack is robust to distribution shift
> Following your suggestion, we run **new synthetic experiments in a setting where there is distribution shift between the data the tokenizer was trained on and the data available to the attacker**. We see that the performance degrades gracefully in this setting, **remaining strong even under extreme shift**. More details are given in the global review.
>
> ## Starting vocabulary and pretokenization
> For BPE tokenizers, the starting vocabulary is fixed at the 256 possible bytes, so we do not need to guess the starting vocabulary to use our attack. Furthermore, all models we study release the pretokenization rules in their tokenizer configuration.
>
> ## Token frequency distribution
> Indeed, in general we cannot expect that every token will be uniformly distributed throughout the corpus. Thus, one would expect a discrepancy between token counts for e.g. one half of the corpus vs halved counts for the full corpus. In our experiments we account for this by subsampling the data, giving some sample to the tokenizer to train and a different sample to the attacker. We show that performance improves as the amount of data given to the attacker increases in Figure 4 of §6.1. This makes sense, as the empirical token frequencies will converge to the true frequencies.
>
> ## Runtime scaling
> As noted previously, it’s difficult to characterize the number of iterations required because it depends heavily on the input data itself. Here is what we observe empirically:
>
> - The pair counting time scales like $O($pair counting data size * number of merges$)$ and doesn't depend on the data.
> - The solving time scales roughly like $O($sqrt(pair counting data size) * (number of merges)$^2)$ but is dependent on the data.
>
> (Note the number of merges is equal to the vocabulary size).

---

> > ### Comment · Reviewer_2xkV · 2024-08-08
> > **Thanks!**
> >
> > I think the authors answered most of my questions. Regarding theoretic analysis part, it is a bit weak, but overall the proposed idea is still novel and this type of attack is very interesting. I will keep my score.

---

### Official Review · Reviewer_rZKP · 2024-07-21

**Soundness:** 3
**Presentation:** 3
**Contribution:** 3
**Rating:** 5
**Confidence:** 3

**Summary:**

The paper "Data Mixture Inference Attack: BPE Tokenizers Reveal Training Data Compositions" presents a significant contribution to the field of machine learning by introducing a novel attack called "data mixture inference." This attack aims to uncover the composition of pretraining data used in language models (LMs), an aspect often kept secret even when the model parameters are open-sourced. By leveraging information from byte-pair encoding (BPE) tokenizers, a common component of modern LMs, the authors successfully infer the proportions of different domains, languages, or code in the training data.

**Strengths:**

1. One of the key strengths of the paper is its demonstration of the attack's effectiveness. Through controlled experiments on known mixtures of natural languages, programming languages, and data sources, the authors show that their attack can recover mixture ratios with high precision, outperforming random guessing by several orders of magnitude. The attack's application to off-the-shelf tokenizers released with commercial LMs reveals new information about their training data composition, underscoring the practical implications of this work.

2. The paper presents an efficient algorithm for solving the data mixture inference problem. Initially, the problem is formulated as a linear program (LP) with a large number of constraints (scaling with the cube of the vocabulary size, O(V^3)). However, the authors introduce several optimizations, such as truncating the merge list, efficiently storing pair counts, and using lazy constraint generation. These optimizations make the problem computationally tractable, allowing for practical application of the attack.

3. The practical implications of this attack are noteworthy. It can potentially allow for the auditing of pretraining data for biases, reveal proprietary information about model construction, and enable targeted data poisoning attacks.

4. The paper is well-written and organized, with the inclusion of an appendix with additional details and results further enhances the paper's clarity and accessibility.

**Weaknesses:**

1. The attack is specifically designed for BPE tokenizers, which, while widely used, are not the only tokenization method. The authors acknowledge this limitation and suggest that other tokenization algorithms might also leak information about training data, but the generalizability of the attack to other tokenization methods is not explored in this work.

2. The attack's reliance on sampling and estimation of pair frequencies could lead to false positives or negatives, especially when dealing with low-resource languages or domains. The paper does not provide a detailed analysis of the attack's robustness to such sampling errors.

**Questions:**

1. The central idea of your paper relies on the fact that the pretraining data was used to train the tokenizer, with the proportions of available data determining the merge rules. However, what happens if a similar domain dataset is used to train the tokenizer with different proportions than those used during pretraining? For instance, if the tokenizer is trained with 40% coding, 40% English, and 20% non-English data, but the pretraining data proportions are 20% coding, 50% English, and 30% non-English, how would this affect the effectiveness and accuracy of your data mixture inference attack? Have you considered or tested such scenarios in your experiments?

2. What are the implications of knowing the composition of the pretraining data? Specifically, what types of attacks could this knowledge facilitate, and what are the general implications of data mixture inference? Why is it important to answer the question of data mixture inference? If your paper addresses these points, please elaborate on how this knowledge could impact the security and integrity of language models. Because by just knowing the broad mixture categories and their weights, it doesn't allow for targeted data poisoning. But it seems it can be a stepping stone for building a sophisticated attacker. Would like to have your thoughts on the same?

**Limitations:**

Yes

---

> ### Author Rebuttal · Authors · 2024-08-07
>
> We thank the reviewer for recognizing our work as a “significant contribution to the field” and its “noteworthy practical implications.” We have taken all of your suggestions into account and hope that our response addresses your concerns.
>
> ## Today’s LLMs use BPE
>
> While it is true that other tokenization algorithms may exist, BPE is the universal choice for today’s production models; in fact, **we manually verified that all top 100 models on ChatbotArena use BPE tokenizers**. In the future, powerful LMs that use tokenizers other than BPE may be released, but we cannot guess what algorithm they will use.
>
> ## Effect of sampling error on attack performance
>
> The linear program we propose is explicitly designed to handle outlier pairs with significantly higher or lower counts than expected using the slack variables $v_p$ and $v^{(t)}$. Of course the actual performance under sampling noise will vary depending on the data, which is why we use sampling in our synthetic experiments, both for the tokenizer’s training data and the data used by the attacker to estimate pair counts. We find that our attack works well, even using these noisy counts. Note that we do have many low-resource languages in our experiments!
>
> We also provide **new results examining the case where the data available to the attacker deviates significantly from that of tokenizer training**. *Please see the global response for details.* In this setting, the counts will deviate from the expected amount due to the distribution shift in addition to the sampling noise. We see that our attack’s performance degrades gracefully in this setting, **remaining strong even under extreme shift**.
>
> ## Further discussion of implications
>
> Since the ideal tokenizer training data is in-distribution with respect to the pretraining data, the tokenizer’s training distribution reflects the model creator’s priorities (see e.g. the recent shift to tokenizers trained on more multilingual data in GPT-4o, Gemma, and Llama-3).
>
> In terms of enabling more specific attacks, data mixture inference gives a useful characterization of the “attack surface” of a model, which could be useful in many ways. For example:
>
> 1. One could leverage discrepancies between the tokenizer training distribution and the pretraining distribution to find “glitch tokens” [1] that trigger unwanted behavior in the target models.
> 2. Knowing the data mixture could help accelerate model stealing attacks, such as the “logprob-free” attacks of [2], since knowing the token distribution gives a useful prior for the (unknown) log-probabilities of each token at generation time.
> 3. When performing membership inference, the data mixture could be used to choose what members to prioritize checking (e.g. we found lots of book data → check for book membership first).
>
> [1] Fishing for Magikarp: Automatically Detecting Under-trained Tokens in Large Language Models. 2024.
>
> [2] Stealing Part of a Production Language Model. 2024.
>
> ## Tokenizer training data vs pretraining data
>
> While pretraining data and tokenizer data are ideally from the same distribution, we do not know whether this is actually the case for the LLMs we study, due to the general opaqueness of model development. Our attack specifically uncovers the *tokenizer* training distribution. We will ensure this is clear in the final version of the paper.

---

> > ### Comment · Reviewer_rZKP · 2024-08-13
> > **Thank you for the response**
> >
> > Thank you authors for the response. After reading the response and other reviewers' comments, I decide to keep the original score unchanged.

---

### Author Rebuttal · Authors · 2024-08-07

Thank you to the reviewers for observing the “novelty” of our data mixture inference attack, its empirical “effectiveness” and the “noteworthy practical implications” of uncovering information about “a fundamental design decision”!

Following insightful reviewer suggestions, we have run several new experiments which we believe strengthen our paper greatly and will be incorporated into the next revision of the paper.

## Our attack beats baseline based on tokenizer performance

Following reviewer suggestions, we provide a **new baseline based on the tokenizer’s encoding efficiency for each language**, and find that **our attack outperforms it by a factor of $10^5$**. In particular, we consider the byte-to-token ratio for the given tokenizer on each language, normalized by that of a tokenizer trained on only that language (with the same total amount of data). The normalization adjusts for any language-specific characteristics that make it inherently easier or harder to encode. We expect that, for instance, if the target language is a *large* proportion of the training data, then it will encode *more* bytes per token (after normalization).

We plot the relationship between the true training proportion and (normalized) tokenizer efficiency in *Figure 1 of the attached PDF*, using the setting where tokenizers are trained on mixtures of $n=10$ languages. As we would expect, the more training data there is for a language, the more efficiently the tokenizer encodes it! However, it is clear that the correlation is not nearly strong enough to precisely recover the true proportion ($x$-axis) precisely given just the encoding efficiency ($y$-axis).

More rigorously, we learn a linear model on the log-log relationship to predict the true proportion given the encoding efficiency. Then we renormalize the predictions for the set of languages into a probability distribution. **This gives $\log_{10}$ MSE of $-2.22$, which is marginally better than random guessing at $-1.84$ but far worse than our attack’s performance at $-7.66$!** Thus, while the baseline gives us a better guess, it does not come close to achieving the kind of precision possible with our attack.

## Our attack is surprisingly robust to unaccounted-for categories

We investigate our attack’s performance when there are unaccounted-for categories. We use the setting with tokenizers trained on mixtures of $n=112$ languages, and omit between 1 and 50 languages when running the solver. To calculate the MSE, we compare our prediction against the renormalized ground truth, after removing the omitted languages. This captures whether we can still accurately recover the distribution of known categories! *See Figure 2 in attached PDF* for the relationship between total probability mass of the omitted languages and MSE. We see that MSE degrades predictably as the omitted probability mass increases.

We find that our attack is surprisingly robust. **Even when $1/10$ of the probability mass is unaccounted for, our attack achieves $\log_{10}$ MSE around $-6$, compared to $-7.69$ with the full set of languages** (recall random guessing here is $-3.82$).

## Attack performance remains strong under distribution shift

We run new experiments where the sample data is not drawn from the same distribution as the tokenizer training data. In particular, we use the Oscar dataset (web data) to train tokenizers on mixtures of languages, but estimate pair frequencies using Wikipedia data for the same languages.

Note that Wikipedia data is significantly different from web data in terms of vocabulary (e.g., more formal words, named entities & dates, Wikipedia headers). **Even under this extreme distribution shift, our attack achieves $\log_{10}$ MSE of $-3.53$**, compared to random guessing at $-1.84$ (100x better than random). We expect the distribution shift of a realistic attack will be much less extreme than this (e.g., using a different corpus of web data).

---

### Decision · Program_Chairs · 2024-09-25

**Decision:**

Accept (poster)

**Comment:**

This paper presents a method for inferring the data mixture a model's tokenizer has been trained on. Specifically, by formulating a linear program based on the BPE merges a good approximation of the tokenizer training corpus can be retrieved. While the tokenizer training corpus might be different from the LLM pretraining corpus, these two are usually a similar/identical mixture, hence providing interesting information about model pretraining.

To the reviewers' and my understanding, this work is novel. It is also interesting, useful, evaluated well, and technically sound. Hence I recommend that this paper is accepted.